# *Midkine* is a dual regulator of wound epidermis development and inflammation during the initiation of limb regeneration

**Stephanie L Tsai[1,2], Clara Baselga-Garriga[1,2], Douglas A Melton[1]***

[1]Department of Stem Cell and Regenerative Biology, Harvard University, Cambridge, United States; [2]Department of Molecular and Cellular Biology, Harvard University, Cambridge, United States

**Abstract** Formation of a specialized wound epidermis is required to initiate salamander limb regeneration. Yet little is known about the roles of the early wound epidermis during the initiation of regeneration and the mechanisms governing its development into the apical epithelial cap (AEC), a signaling structure necessary for outgrowth and patterning of the regenerate. Here, we elucidate the functions of the early wound epidermis, and further reveal *midkine* (*mk*) as a dual regulator of both AEC development and inflammation during the initiation of axolotl limb regeneration. Through loss- and gain-of-function experiments, we demonstrate that *mk* acts as both a critical survival signal to control the expansion and function of the early wound epidermis and an anti-inflammatory cytokine to resolve early injury-induced inflammation. Altogether, these findings unveil one of the first identified regulators of AEC development and provide fundamental insights into early wound epidermis function, development, and the initiation of limb regeneration.

## Introduction

Salamanders possess the impressive ability to regenerate their limbs with high fidelity throughout their lifespan (*Tanaka, 2016*; *Flowers et al., 2017*). While most mammals respond to traumatic limb injuries including amputation with a fibrotic scarring response, salamanders instead form a blastema, a transient cellular structure comprised of progenitors derived from different non-epithelial tissues (muscle, connective tissue etc.) that eventually reconstructs the new limb (*Kragl et al., 2009*; *Sandoval-Guzmán et al., 2014*; *Fei et al., 2017*; *Gerber et al., 2018*). Determining the mechanisms required for blastema formation has the potential to provide valuable insights into improving repair and regenerative outcomes in less regenerative-species including humans.

Classical salamander limb regenerative experiments determined that initiation of blastema formation requires the presence of a wound epidermis for both formation and maintenance of the blastema (*Goss, 1956a*; *Thornton, 1957*; *Thornton, 1958*; *Mescher, 1976*; *Tassava and Garling, 1979*). Immediately post-amputation, epithelial cells from the intact peripheral skin migrate radially over the exposed stump tissues and form a thin wound epithelium covering the amputation plane within 12 hours (*Hay and Fischman, 1961*). Unlike normal full thickness skin that contains two layers (epidermis and dermis), the wound epidermis only contains an epithelial layer comprised primarily of keratinocytes, allowing for direct communication between the wound epidermis and underlying stump tissues (*Repesh and Oberpriller, 1978*). This thin wound epidermis persists through the initial wound healing stages of regeneration. As the blastema forms during later stages of regeneration, the early wound epithelium gradually expands into a stratified epidermis commonly called the apical epithelial cap (AEC), which is a signaling structure that maintains blastemal outgrowth and directs patterning (*Boilly and Albert, 1990*; *Christensen and Tassava, 2000*; *Han et al., 2001*; *Christensen et al., 2002*). This expansion occurs primarily via proliferation and subsequent distal

**\*For correspondence:**
dmelton@harvard.edu

**Competing interests:** The authors declare that no competing interests exist.

migration of epithelial cells in the peripheral skin (*Hay and Fischman, 1961*; *Campbell and Crews, 2008*). Classical studies showed that inhibiting early wound epidermis formation by either suturing or grafting full thickness skin over the amputation plane (*Mescher, 1976*; *Tassava and Garling, 1979*) or immediately inserting the amputated limb into the body cavity (*Goss, 1956a*) blocked blastema formation. However, if the amputated limb was inserted into the body cavity after the wound epidermis formed, regeneration proceeded normally, demonstrating an early need for the wound epithelium (*Goss, 1956b*). Furthermore, preventing AEC development by either continuous removal of the wound epidermis (*Thornton, 1957*) or daily irradiation to inhibit proliferative expansion also hindered limb regeneration (*Thornton, 1958*). Together, these findings established the necessity for a wound epidermis/AEC throughout different stages of limb regeneration.

The nature of wound epidermis dependence has been more well explored during later blastemal stages of regeneration. The AEC expresses mitogens and patterning factors that help to maintain blastemal cell proliferation and pattern the regenerate (*Boilly and Albert, 1990*; *Campbell and Crews, 2008*; *Ghosh et al., 2008*; *Satoh et al., 2008*). However, the roles of the wound epidermis during early stages of regeneration are less clear. Moreover, the early wound epidermis is likely to be functionally distinct from the AEC as the wound healing micro-environment differs substantially from that of the blastema. Preventing early wound epidermis formation was shown to affect the maintenance, but not the onset of proliferation of progenitors (*Mescher, 1976*; *Tassava and Garling, 1979*; *Johnson et al., 2018*), though it is worth noting that the early wound epidermis does express molecules capable of inducing proliferation (*Sugiura et al., 2016*). Additionally, some dedifferentiation of muscle, bone and cartilage tissue was still observed when wound epidermis formation was prevented, although not to the same extent as normal regenerating limbs. Altogether these studies indicated that the early wound epidermis is involved in maintaining progenitor cell cycling and may also play a role in tissue histolysis.

More recently, bulk and single cell transcriptional limb regeneration studies have further characterized gene expression in the early wound epidermis and AEC (*Campbell et al., 2011*; *Leigh et al., 2018*; *Tsai et al., 2019*). Yet, it remains unknown what gene expression programs in regenerating tissues depend on the presence of the early wound epidermis. The initiation of blastema formation is a complex event with inflammation, tissue histolysis, extracellular matrix (ECM) remodeling, and cell cycle re-entry of progenitors all transpiring concurrently (*McCusker et al., 2015*). Aside from the induction of proliferation, whether and to what extent the early wound epidermis may regulate these other processes remains relatively unexplored. Furthermore, little is known about the molecular mechanisms controlling the transition from the early wound epithelium into the AEC, a major milestone for successful limb regeneration.

To investigate these questions, we sought to deconvolute the functional roles of the wound epidermis from the rest of the regenerating limb tissues by revisiting the classical full skin flap surgical model to inhibit wound epidermis formation during limb regeneration. Using modern-day transcriptional approaches, we illuminate the roles of the early wound epidermis and identify distinct wound epidermis-dependent transcriptional programs in dividing progenitors and the surrounding regenerating tissues. In addition to the discovery of putative regulators of wound epidermis formation and function, our approach enabled us to identify *midkine* (*mk*) as a regulator of both AEC development and early inflammation, key events during the initiation of blastema formation. Altogether, these findings provide important insight into the molecular basis for wound epidermis dependence and further advance our understanding of wound epithelium/AEC biology during limb regeneration.

## Results

### The wound epithelium is a major regulator of inflammation, ECM remodeling, and tissue histolysis during early stages of regeneration

We previously developed a method leveraging differential DNA content during the cell cycle to enrich for dividing progenitors during early stages of regeneration (*Tsai et al., 2019*), facilitating analysis of the distinct transcriptional programs active in blastemal progenitors apart from those of the surrounding tissues (non-dividing cells in stump tissues and the wound epidermis). To examine the functional roles of the early wound epidermis, we asked how the transcriptional programs of these three populations changed when wound epidermis formation was prevented. To do this, we

utilized the classical full skin flap (FSF) surgical model pioneered by Anthony Mescher (*Mescher, 1976*). We performed full skin flap surgeries on regenerating axolotl limbs to prevent epithelial cell migration and contact with exposed stump tissues, inhibiting both wound epidermis formation and limb regeneration (*Figure 1—figure supplement 1A–B*). Since preventing early wound epidermis formation via full skin flap surgery does not affect induction of proliferation (*Mescher, 1976*; *Tassava and Garling, 1979*; *Johnson et al., 2018*), we applied our method here to profile blastemal progenitors and the surrounding regenerating tissues in full skin flap sutured limbs at 5 days post-amputation (dpa), around the onset of proliferation. By leveraging the data generated from our previous analysis (*Tsai et al., 2019*), we were able to compare the gene expression programs in full skin flap sutured regenerating limbs to both intact limbs and normal regenerating limbs from the same time point (*Figure 1A*, *Figure 1—figure supplement 1C*). This approach allowed us to identify wound epidermis-dependent gene expression.

Preventing wound epidermis formation led to aberrant growth factor signaling pathway activation/repression in regenerating stump tissues (*Figure 1B*). Pathway analysis revealed that while non-dividing cells exhibited no major differences, dividing cells (i.e. enriching for progenitors) exhibited a

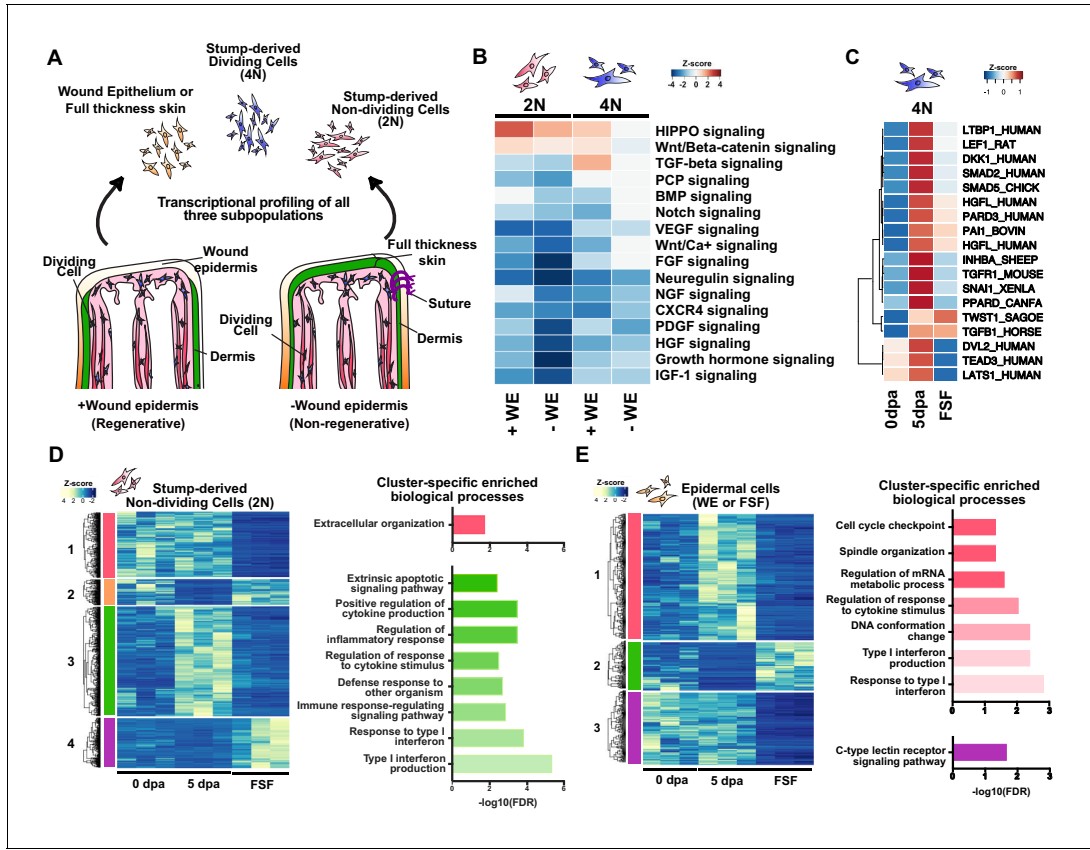

**Figure 1.** The early wound epidermis modulates inflammation, ECM remodeling, and tissue histolysis. (A) Design of transcriptional profiling experiment. (B) Heatmap of Ingenuity Pathway Analysis (IPA) Z-score predictions for changes in signaling pathways in injured non-epithelial stump tissues (2N and 4N) in regenerative and full skin flap conditions. (C) Heatmap of normalized transcript levels of components/regulators of the canonical Wnt, TGF-beta, and HIPPO signaling pathways in dividing cells (4N). (D–E) Heatmaps and cluster-specific enriched biological processes of differentially expressed annotated transcripts in stump-derived non-dividing cells (2N) (D) and epithelial cells of the full skin flap (E) in regenerating vs. full skin flap conditions reveals global dysregulation of inflammation, ECM regulation, and tissue histolysis. Differentially expressed transcripts can be found in *Supplementary files 1–3*. Representative histological images of full skin flap sutured vs. normal regenerating limbs, PCA analysis of the samples, as well as enriched biological processes and related differentially expressed transcripts are shown in *Figure 1—figure supplement 1*. Z-score predictions shown in B from the IPA analysis are available in *Figure 1—source data 1*.

The online version of this article includes the following source data and figure supplement(s) for figure 1:

**Source data 1.** Z-score predictions from the IPA analysis (for panel *Figure 1B*).

**Figure supplement 1.** Sequencing of full skin flap sutured limbs reveals global dysregulation of ECM regulation, inflammation, and tissue histolysis.

loss in predicted activation of canonical Wnt, HIPPO, and TGF-beta signaling. Many components of these pathways were also expressed at lower levels (*Figure 1C*), indicating that the wound epidermis induces transcriptional activation of these pathways in dividing progenitors. Interestingly, some components of these pathways including *tgfb1* and *twist1* were not transcriptionally affected, suggesting that their expression in progenitors is wound epidermis-independent. Notably, the overall transcriptional profiles of dividing progenitors in both cases were relatively similar (596 differentially expressed transcripts, 313 annotated) (*Supplementary file 1*), signifying that while the wound epidermis activates key signaling pathways, the overall gene expression programs in early progenitors are largely wound epidermis-independent. This lends transcriptional evidence to reinforce the notion that the earliest transcriptional programs of progenitors are likely driven in response to the injury itself (*Tassava and Loyd, 1977*; *Wagner et al., 2017*; *Johnson et al., 2018*).

In contrast to the dividing progenitors, the transcriptional programs of the surrounding tissues diverged considerably. We found that 3911 transcripts (982 annotated) were differentially expressed in non-dividing cells in regenerating stump tissues (*Figure 1D*, *Figure 1—figure supplement 1D–E*, *Supplementary file 2*). The majority of these transcripts pertained to genes involved in ECM regulation, inflammation, and tissue histolysis. Many ECM-components and regulators that maintained expression in intact and normal regenerating tissues were aberrantly down-regulated (e.g. *fbn2*, *finc*, *has2*), and regenerative ECM components as well as degrading enzymes (e.g. *tenascin*, *mmp2*, and *mmp13*) failed to be induced, indicating defects in ECM remodeling. Furthermore, there was a high representation of mis-regulated immune-related transcripts, both pertaining to pro- and anti-inflammatory processes. In particular, components of pro-inflammatory signaling pathways including toll-like receptor (TLR) signaling and interferon α/β/γ signaling, as well as anti-inflammatory pathways including PPARα/RXRα signaling and *anxa1* signaling were dysregulated (*Figure 1—figure supplement 1D–E*). Enzymes involved in tissue histolysis were mis-expressed as well. Notably, expression of the major bone matrix degrading enzyme *catk* failed to be induced, while the major bone and cartilage matrix component *col1a1* maintained expression, indicating potential defects in bone resorption, a process that begins during the first week of regeneration (*Fischman and Hay, 1962*). Altogether, these data suggest that the wound epidermis is a major regulator of inflammation, ECM remodeling, and tissue histolysis in regenerating stump tissues during early stages of limb regeneration.

The presence of the dermis in full skin flap sutured limbs additionally presented a crude, yet effective opportunity to identify epithelial transcriptional programs dependent on direct contact between the wound epidermis and regenerating stump tissues. We found that 1060 transcripts (480 annotated) were differentially expressed (*Figure 1E*, *Figure 1—figure supplement 1F*, *Supplementary file 3*). Similar to the non-dividing cells in regenerating stump tissues, many transcripts involved in ECM regulation and inflammatory signaling (both pro- and anti-inflammatory) were also dysregulated in full skin flap epithelial cells (*Figure 1—figure supplement 1F*). Components of interferon and cytokine signaling failed to be induced in full skin flap derived epithelial cells, while anti-pathogenic mucins including *muc5a* and *muc5b* were overexpressed, suggesting dysregulation of inflammatory processes. Interestingly, *mpeg1* commonly failed to be induced in the full skin flap epithelium as well as non-dividing cells in regenerating stump tissues, potentially indicating a defect in macrophage infiltration during early regeneration. Furthermore, ECM components expressed in intact skin (e.g. *otoan*, *k1c15*, *k1c19*, *k2c5*) lost expression in full skin flap epithelia, and other ECM regulators including *finc*, *palld*, and *mmp3* were also differentially expressed. These data strongly suggest that direct communication between regenerating mesenchymal tissues and epithelial cells is crucial for proper modulation of early inflammatory responses and ECM remodeling.

## Midkine is highly expressed in the basal layers of the wound epidermis/AEC and blastemal progenitors

*Midkine* (*mk*) is a pleiotropic growth factor cytokine that has been shown to regulate a variety of processes including development, tumorigenesis, inflammation, and tissue repair (*Muramatsu, 2010*; *Weckbach et al., 2011*; *Gramage et al., 2015*; *Karaman and Alitalo, 2017*). Interestingly, we noticed that *mk* expression was induced in the wound epidermis and highly upregulated in early regenerating stump tissues. In full skin flap sutured limbs, *mk* expression decreased across all transcriptionally profiled subpopulations (*Figure 2—figure supplement 1A*). Specifically, while *mk* expression in regenerating stump tissues was lower, *mk* failed to be transcriptionally induced in full

skin flap epithelial cells. These data suggested that *mk* may modulate wound epidermis development/function. Hence, we chose to further explore the role of *mk* during limb regeneration.

To determine when and in which cell types *mk* is expressed during limb regeneration, we performed a series of single and double in situ hybridizations. Time course in situ hybridization of *mk* confirmed that it is highly expressed during limb regeneration, but not in full skin flap sutured regenerating limbs (*Figure 2A–G*). In non-regenerating intact limbs, *mk* was lowly expressed in cells interspersed throughout the muscle and dermis (*Figure 2A–A'*). During regeneration, *mk* became strongly expressed in both the basal layers of the wound epidermis and mesenchymal cells within regenerating stump tissues beginning at 3 dpa (*Figure 2B–C'*). To determine whether *mk* expression was first up-regulated in regenerating stump tissues or the wound epidermis, we performed a more

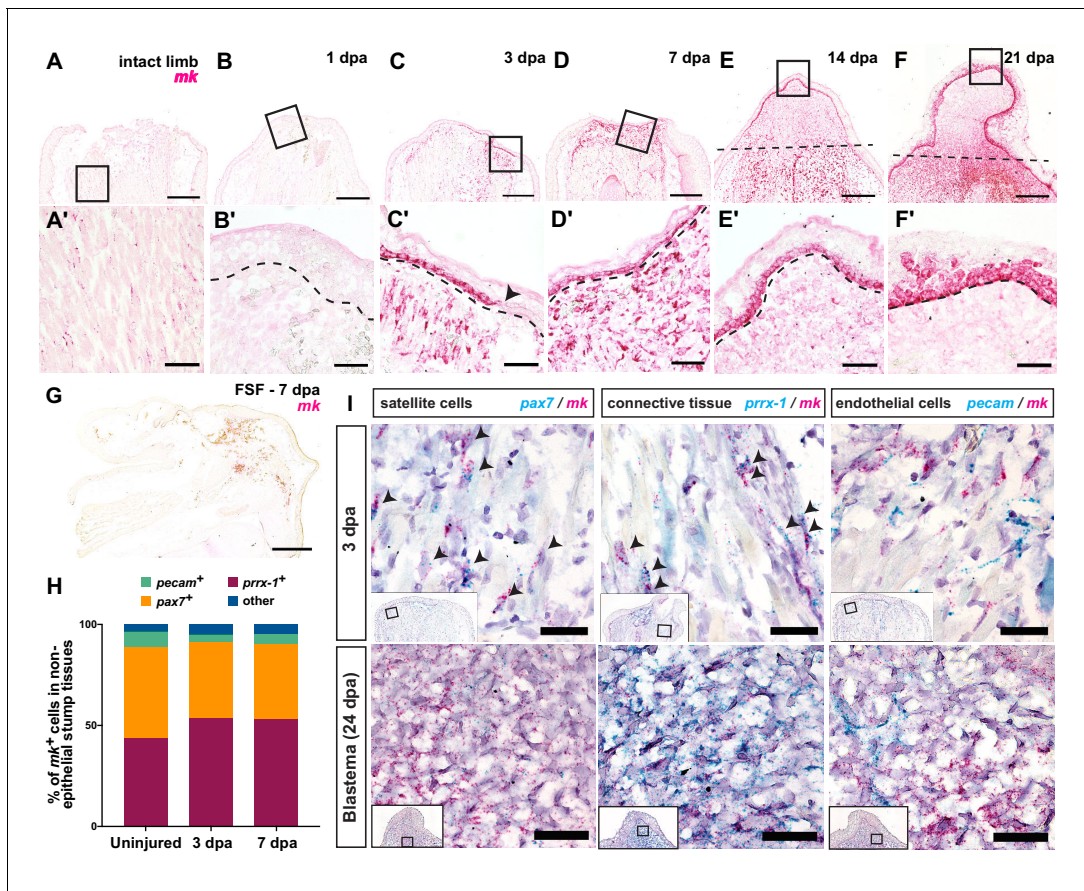

**Figure 2.** *Midkine* (*mk*) is highly expressed in the basal layers of the wound epidermis/AEC and blastemal progenitors. (**A–F'**) Timecourse RNAscope in situ hybridization of *midkine* at 0 (intact), 1, 3, 7, 14, and 21 dpa. Insets in A-F are shown in A'-F' at higher magnification. Arrowhead in C' denotes the beginning of the wound epidermis. Dotted line marks amputation plane in E and F or wound epidermis/AEC boundary in B'-F'. (**G**) In situ hybridization of *mk* in full skin flap sutured limbs at 7 dpa. Axolotl MK protein expression can be found in *Figure 2—figure supplement 1*. (**H**) Breakdown of the percentages of *mk*+ cells that are *pax7*+, *prrx-1*+, and *pecam*+ in regenerating stump tissues during early stages of regeneration. At 3 dpa, a total N of 1579, 444, and 1180 cells were counted for *pax7*, *prrx*-1, and *pecam* double in situs, respectively. At 7 dpa, a total N of 456, 1043, and 274 cells were counted for *pax7*, *prrx*-1, and *pecam* double in situs, respectively (*Figure 2—source data 1*). (**I**) Representative images of RNAscope double in situ hybridization of *mk* with *pax7* (left), *prrx*-1 (middle), or *pecam* (right) at early (3 dpa) and later blastema (24 dpa) stages. Insets depict where higher magnification images were taken. Black arrowheads mark double positive cells. More detailed analysis of the onset of *mk* expression in early stages of regeneration, representative images of double in situ hybridization of *mk* with cell type-specific markers in uninjured tissue, as well as the analysis of *mk* co-expression with the monocyte marker *csf1r* can be found in *Figure 2—figure supplement 2*. Scale bars, A-G: 500 μm, A'-F': 100 μm, I: 50 μm. dpa, days post-amputation, FSF, full skin flap.

The online version of this article includes the following source data and figure supplement(s) for figure 2:

**Source data 1.** Raw counts for *mk* double in situ hybridizations with cell-type specific markers.

**Figure supplement 1.** MK protein is found throughout the wound epidermis/AEC.

**Figure supplement 2.** *Mk* is expressed in satellite cells, connective tissue, and endothelial cells in intact tissues.

detailed analysis of the onset of *mk* expression prior to 3 dpa. In situ hybridization of *mk* at 24 and 56 hours post-amputation (hpa) revealed a small population of *mk*⁺ cells interspersed throughout regenerating stump tissues with little to no expression of *mk* in the basal layers of the wound epidermis until 72 hpa (*Figure 2—figure supplement 2A*). Together, these data indicate that *mk* is first expressed in regenerating stump tissues prior to the induction of expression in the wound epidermis. During later stages of regeneration, *mk* expression persisted at high levels in the basal layers of the AEC and lower levels in the blastema (*Figure 2D–F'*). Interestingly, we also noticed relatively strong *mk* expression within the non-regenerating tissue at the base of the blastema (Figure F-F'), suggesting *mk* may play a role in differentiation during later stages of regeneration. Immunostaining of axolotl MK protein further revealed the presence of MK protein throughout the AEC and blastema, providing evidence that it is indeed secreted (*Figure 2—figure supplement 1B–F'*).

Next, to identify which cell types expressed *mk* in regenerating stump tissues, we performed double in situ hybridization of *mk* with various cell type-specific markers in non-regenerating and regenerating limbs. Because we had observed *mk*⁺ cells interspersed in the muscle and dermis of intact tissues, we surmised that *mk* may be expressed at low homeostatic levels in subsets of connective tissue, muscle, or endothelial cells. To examine this possibility, we performed double in situ hybridization of *mk* with *prrx-1*, *pax7*, and *pecam*, which are markers for connective tissue cells (*Gerber et al., 2018*), muscle satellite cells (*Sandoval-Guzmán et al., 2014*), and endothelial cells (*Whited et al., 2013*), respectively. Indeed, our analysis revealed that approximately 96.4% of *mk*⁺ expressing cells in non-regenerating tissues were comprised of *prrx-1*⁺ connective tissue cells (~43.8% of *mk*⁺ cells), *pax7*⁺ satellite cells (~44.9% of *mk*⁺ cells), and *pecam*⁺ endothelial cells (~7.7% of *mk*⁺ cells) (*Figure 2H*, *Figure 2—figure supplement 2B*).

Interestingly, we observed a relatively similar breakdown of *mk*⁺ cells across all three cell types during early stages limb regeneration (*Figure 2H–I*). At both 3 and 7 dpa, *mk*-expressing cells could be divided into connective tissue cells (~53–54%), satellite cells (~36–37%), and a small percentage of endothelial cells (4–5%). Given heavy immune cell infiltration during wound healing stages of regeneration, we also investigated whether *mk* was expressed in monocytes by performing double in situ hybridization with *csf1-r*, a monocyte marker (*Grayfer et al., 2014*). We found that only a small percentage (~9%) of *mk*⁺ cells were *csf1-r*⁺ and only 3.7% of *csf1-r*⁺ monocytes were *mk*⁺ at 5 dpa (*Figure 2—figure supplement 2C–D*), indicating *mk* is not predominantly expressed in monocytes. In addition, while the densely packed cells within the blastema at 24 dpa precluded the accurate quantification of double positive cells, we did observe overlapping expression patterns between *mk* and both *prrx-1* and *pax7*, but not *pecam* (*Figure 2I*), indicating *mk* expression persists in connective tissue and muscle-derived blastemal cells. Altogether, these data strongly indicated that *mk* expression is highly up-regulated during limb regeneration in the majority of blastemal cells, since satellite and connective tissue cells comprise most of the blastema (*Fei et al., 2017*; *Gerber et al., 2018*). As communication between blastemal cells and the basal layers of the wound epidermis/AEC is crucial for blastema formation (*Stocum and Dearlove, 1972*), the unique expression pattern of *mk* in both cell types suggested that *mk* may play an important role in facilitating signaling between the two tissues.

## Chemical and genetic perturbation of mk impairs limb regeneration

We next examined whether *mk* loss-of-function affected limb regeneration using both chemical and genetic methods. We first performed drug treatments with a small molecule inhibitor of MK, iMDK (*Masui et al., 2016*) (*Figure 3*). Treatment with iMDK beginning at either 0 or 7 dpa completely inhibited limb regeneration (*Figure 3A–B*). Alcian blue staining of limbs at 60 dpa confirmed that no skeletal structures had regenerated in iMDK-treated limbs (*Figure 3C–D*). We also generated F0 *mk* CRISPR mutants to perturb *mk* genetically (*Figure 3E*). The slow development of axolotl embryos permits high penetrance of mutations allowing phenotyping to occur at the F0 stage (*Fei et al., 2018*). *Mk* F0 mutants generated by targeting the start codon with CRISPR technology (see Materials and methods) developed with no visible abnormalities and anatomically normal limbs (*Figure 3—figure supplement 1*). Many of these mutants had mosaic null genotypes. Upon amputation, *mk* F0 mutants exhibited delayed blastema formation and regeneration (*Figure 3E–F*). Notably, measurements of blastema length at 14 dpa revealed that the severity of delay in regeneration segregated based on genotype, with *mk* mosaic null animals (>99% mutant alleles) exhibiting the most

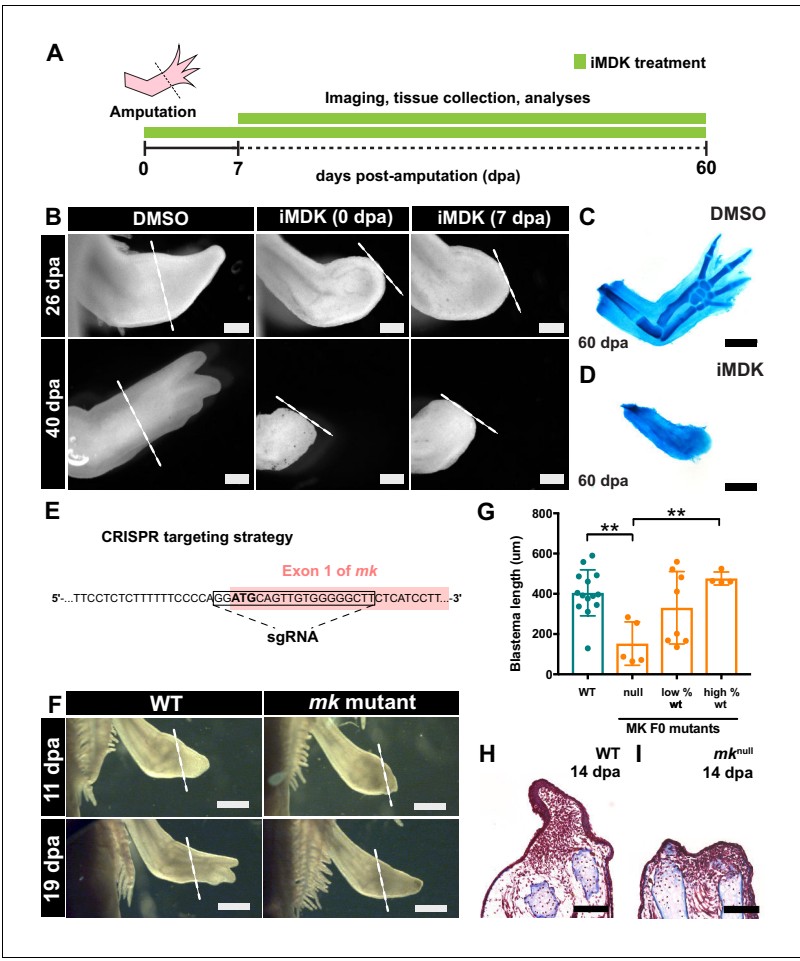

**Figure 3.** Chemical and genetic perturbations of *mk* impair limb regeneration. (A) Experimental design. (B) Brightfield images of DMSO- or iMDK-treated limbs (N = 4/4 iMDK-treated in each condition did not regenerate). (C–D) Alcian blue staining of DMSO- or iMDK-treated limbs at 60 dpa. (E) CRISPR strategy to target the start codon of *mk* to generate mutants. Control animals were generated using a non-targeted tracrRNA/cas9 complex and were unmodified at the *mk* locus. (F) Brightfield images of regenerating limbs from control animals or *mk* F0 mutants. (G) Quantification of blastema length at 14 dpa in control or *mk* mutant limbs. The severity of the delay in regeneration segregated based on genotype of the animal as either control ($mk^{WT}$), mosaic null ($mk^{null}$), or partially modified (low or high % WT alleles) (N = 14 control $mk^{WT}$, 5 $mk^{null}$, 8 low % WT, 4 high % WT). Graph is mean ± SD. (H–I) Representative images of picro-mallory stained sections of regenerating limbs in control animals or *mk* null mutants. Example genotyping analyses and *mk* null mutant immunofluorescence validation can be found in *Figure 3—figure supplement 1*. **$p<0.005$, two-tailed unpaired Student's t-test was employed. Each N represents one limb from a different animal. White dotted lines mark amputation plane. Scale bars, B-D, H-I: 500 µm, F: 1 mm. WT, wildtype, dpa, days post-amputation.

The online version of this article includes the following figure supplement(s) for figure 3:

**Figure supplement 1.** CRISPR generation of *mk* mosaic null mutants.

---

severe delay compared to control animals (*Figure 3G–I*). These data demonstrate that both chemical and genetic perturbations of *mk* impair limb regeneration.

## Mk loss-of-function leads to impaired AEC development and persistent inflammation

To elucidate the cellular mechanism of the regeneration defect, we further characterized both iMDK-treated and *mk* null ($mk^{null}$) mutant regenerating limbs (*Figure 4*). Interestingly, iMDK-treated regenerating limbs had significantly lower percentages of total EdU+ cells during late (14 dpa), but not early stages of limb regeneration (5 dpa) (*Figure 4A–B*). Furthermore, blastemal cells accumulated

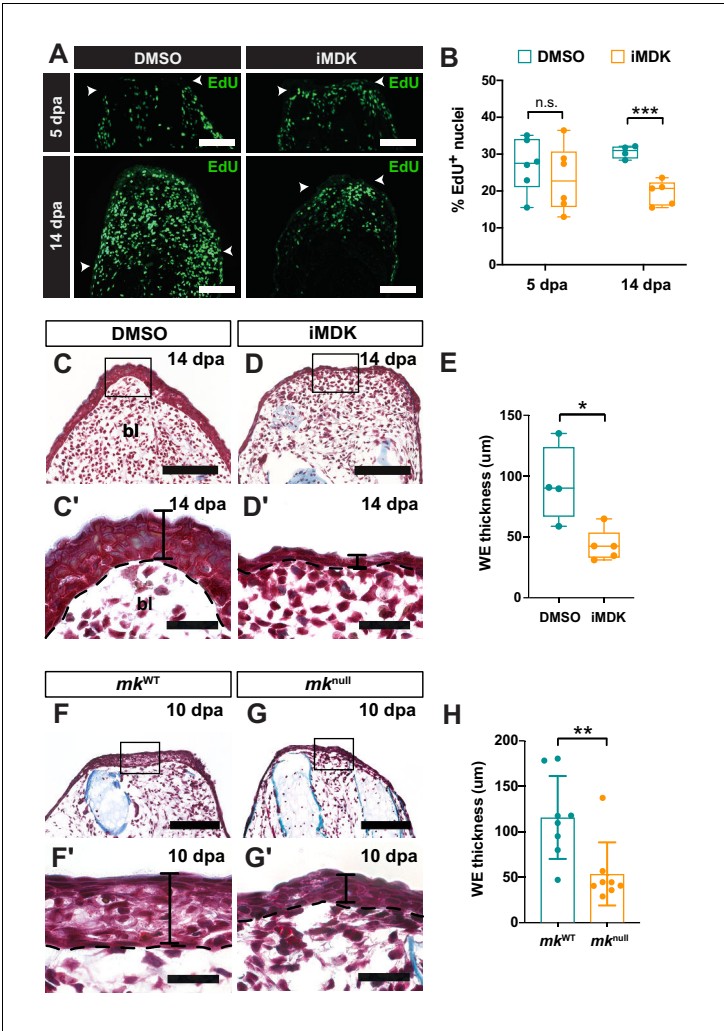

**Figure 4.** *Mk* perturbations result in defective AEC development. (**A–B**) EdU staining (**A**) and quantification (**B**) of the total percentage of EdU⁺ nuclei of DMSO/iMDK-treated limbs at 5 dpa (N = 6 DMSO, 6 iMDK) and 14 dpa (N = 4 DMSO, 5 iMDK). White arrowheads denote the amputation plane in A. (**C–D'**) Picro-mallory stained sections from regenerating limbs from DMSO/iMDK-treated animals. Higher magnification images of the AEC are shown in C'-D'. (**E**) Quantification of wound epidermis (WE) thickness in DMSO/iMDK-treated regenerating limbs at 14 dpa (N = 4 DMSO, 5 iMDK). (**F–G'**) Picro-mallory stained sections from regenerating limbs from regenerating limbs of control WT (*mk*ᵂᵀ) or *mk*ⁿᵘˡˡ animals. Higher magnification images of the developing AEC at 10 dpa are shown in F'-G'. (**H**) Quantification of WE thickness in regenerating limbs of *mk*ᵂᵀ or *mk*ⁿᵘˡˡ animals at 14 dpa (N = 8 *mk*ᵂᵀ, 8 *mk*ⁿᵘˡˡ). Picro-mallory staining of a 40 dpa iMDK-treated limb is shown in *Figure 4—figure supplement 1*. Each N represents a limb from a different animal. Two-tailed unpaired student's t-test was used for statistical analysis. Graphs are mean ± SD. *p<0.05, **p<0.005, ***p<0.001. Scale bars, A, C-D, F-G: 200 μm, C'-D', F'-G': 100 μm. bl, blastema, dpa, days post-amputation.

The online version of this article includes the following figure supplement(s) for figure 4:

**Figure supplement 1.** iMDK-treated limbs do not regenerate.

at the amputation plane in iMDK-treated limbs even at 40 dpa (*Figure 4—figure supplement 1*), however blastemal outgrowth never occurred, altogether suggesting that *mk* is involved in maintenance, but not induction of cellular proliferation. In addition, we observed a significantly thinner wound epidermis in both iMDK-treated and *mk*ⁿᵘˡˡ mutant limbs during later stages of regeneration compared to controls (*Figure 4C–H*). These dual observations indicated a defect in wound epidermis expansion and function during AEC development.

Further analysis of TUNEL staining in both *mk*<sup>null</sup> mutant and iMDK-treated regenerating limbs revealed that the defect in AEC development was primarily attributed to higher levels of cell death in the wound epidermis in both contexts (*Figure 5*). Altogether, these data indicate that *mk* mainly regulates early wound epidermis expansion by acting as a survival factor during the early wound epidermis transition to the AEC. In addition, while iMDK-treated limbs exhibited persistently high levels of cell death in blastemal cells, *mk*<sup>null</sup> mutants had higher levels of cell death during early stages that eventually resolved to that of wildtype limbs, suggesting *mk* is not required for blastemal cell survival (*Figure 5—figure supplement 1*).

To determine the extent of defective AEC development, we also examined other aspects of the wound epidermis including innervation, which has been shown to be important for the AEC transition (*Satoh et al., 2008*; *Satoh et al., 2012*). Interestingly, iMDK-treatment did not impact wound epidermis innervation (*Figure 6A*) and data from a published microarray dataset showed similar induction of *mk* expression in innervated vs. denervated regenerating limbs (*Monaghan et al., 2009*). These results collectively indicate that *mk* operates in a nerve-independent manner. However, while expression of the known wound epidermis/AEC marker WE3 was observed (*Figure 6A*), sequencing of DMSO- and iMDK-treated limbs at 11 dpa revealed transcriptional dysregulation of many wound epidermis genes, suggestive of impaired functionality (*Figure 6B–C*, *Supplementary file 4*) (*Campbell et al., 2011*; *Gerber et al., 2018*; *Leigh et al., 2018*).

Notably, many pro-inflammatory pathways were enriched in inhibitor treated limbs (*Figure 6D–E*) and both *mk*<sup>null</sup> mutant and iMDK-treated regenerating limbs exhibited higher monocyte densities

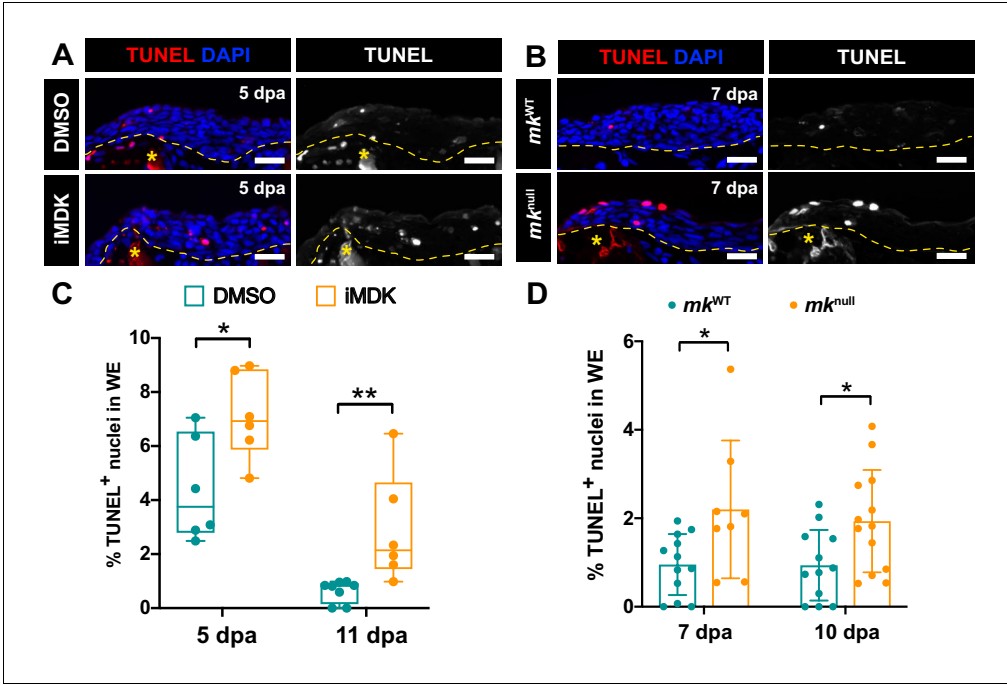

**Figure 5.** *Mk* acts as a critical survival factor during AEC development. (**A**) Representative TUNEL-stained sections from limbs of DMSO and iMDK-treated limbs. (**B**) Representative TUNEL-stained sections from limbs of regenerating *mk*<sup>WT</sup> control or *mk*<sup>null</sup> mutants. (**C**) Quantification of the % TUNEL<sup>+</sup> nuclei in the wound epidermis of DMSO or iMDK-treated limbs limbs at 5 dpa (N = 6 DMSO, 6 iMDK) or 11 dpa (N = 8 DMSO, 6 iMDK). (**D**) Quantification of the percentage of TUNEL<sup>+</sup> nuclei in *mk*<sup>WT</sup> control and *mk*<sup>null</sup> mutants at 7 dpa (N = 12 *mk*<sup>WT</sup>, 8 *mk*<sup>null</sup>) and 10 dpa (N = 12 *mk*<sup>WT</sup>, 13 *mk*<sup>null</sup>). Asterisks mark auto-fluorescent bone. Each N represents a limb from a different animal. Quantification of levels of blastemal cell death in DMSO/iMDK-treated and *mk*<sup>WT</sup>/*mk*<sup>null</sup> regenerating limbs can be found in *Figure 5—figure supplement 1*. Two-tailed unpaired student's t-tests were used for statistical analysis. Graphs are mean ± SD. *p<0.05, **p<0.005. Scale bars: 100 µm. dpa, days post-amputation.

The online version of this article includes the following figure supplement(s) for figure 5:

**Figure supplement 1.** *Mk* is not required for blastemal cell survival.

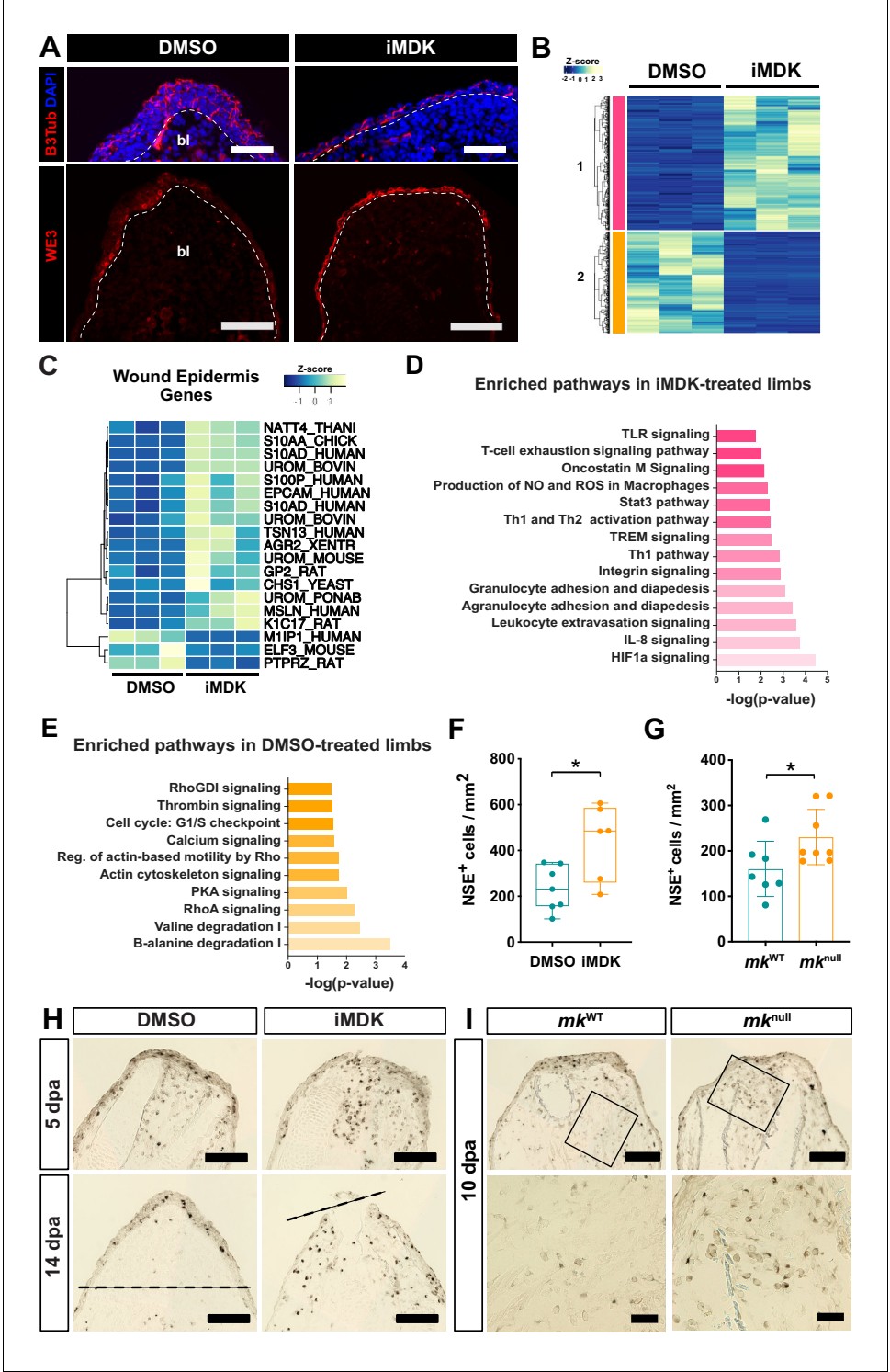

**Figure 6.** iMDK-treated and *mk* mutant regenerating limbs display dysregulated wound epidermis gene expression and persistent inflammation. (**A**) Beta-III tubulin staining or WE3 staining of DMSO- or iMDK-treated limbs. White dotted line marks boundary of wound epidermis/AEC-blastema. (**B**) Heatmap of annotated differentially expressed transcripts in DMSO- and iMDK-treated limbs (N = 3 each) reveals two main clusters (colored pink and orange) of transcripts either enriched in DMSO or iMDK treatments. Transcript expression was normalized per row and plotted as a Z-score. Differentially expressed transcripts can be found in ***Supplementary file 4***. (**C**) Heatmap of normalized TPM expression levels of wound epidermis genes in DMSO- or iMDK-treated regenerating limbs at 11 dpa. (**D**) Plot of enriched pathways in iMDK-treated limbs. (**E**) Plot of

*Figure 6 continued on next page*

*Figure 6 continued*

enriched pathways in DMSO-treated limbs. (**F**) Quantification of NSE$^+$ monocytes at 5 dpa in DMSO/iMDK-treated limbs at 5 dpa (N = 7 DMSO, 6 iMDK). (**G**) Quantification of the density of NSE$^+$ monocytes in $mk^{WT}$ control and $mk^{null}$ regenerating limbs at 10 dpa (N = 7 $mk^{WT}$, 8 $mk^{null}$). (**H**) NSE staining of DMSO- and iMDK-treated limbs at 5 dpa and 14 dpa. Dotted lines demarcate the amputation plane. (**I**) Representative NSE stained sections from regenerating limbs of $mk^{WT}$ control and $mk^{null}$ mutants at 10 dpa. Higher magnification insets are shown in bottom two panels. Each N represents a limb from a different animal. Data demonstrating rescue of $mk^{null}$ phenotypes via overexpression of $mk$ during regeneration in mutant limbs as well as electroporation efficiency metrics can be found in *Figure 6—figure supplements 1*, *2* and *3*. A two-tailed unpaired student's t-test was used for statistical analysis. Graphs are mean ± SD. *p<0.05. Scale bars, A (top): 100 μm, A (bottom), H-I (top): 200 μm, I (bottom): 50 μm. bl, blastema, dpa, days post-amputation.

The online version of this article includes the following figure supplement(s) for figure 6:

**Figure supplement 1.** *Mk* overexpression in $mk^{null}$ regenerating limbs rescues delayed regeneration.

**Figure supplement 2.** *Mk* overexpression in $mk^{null}$ regenerating limbs rescues mutant wound epidermis and monocyte density phenotypes.

**Figure supplement 3.** Electroporation efficiencies are similar between pCAG-tdTomato and pCAG-MK injected $mk^{null}$ mutant and $mk^{WT}$ regenerating limbs.

(*Figure 6F–I*), suggesting that functional perturbation of *mk* likely leads to prolonged inflammation. As we uncovered an immunomodulatory role for the early wound epidermis from our transcriptional data, it is possible that the persistent inflammation in *mk* loss-of-function contexts could be indirectly caused by impaired wound epidermis functionality from defective AEC development.

Most importantly, overexpression of *mk* in $mk^{null}$ mutant regenerating limbs rescued the delayed blastema formation and regeneration phenotype as well as the wound epidermis defects and high monocyte density (*Figure 6—figure supplements 1–3*), confirming that *mk* is indeed a regulator of both AEC development and the resolution of inflammation. As electroporation of constructs into limb epidermis is challenging and only occurs at a very low efficiency (*Fei et al., 2016*), the majority of *mk* was expressed and secreted from regenerating stump tissues. Therefore, these data further suggested that *mk* did not necessarily need to be produced in the wound epidermis to rescue AEC development.

## Mk is sufficient to control proliferation and inflammation in regenerating limbs

As the AEC forms via epithelial cell proliferation in peripheral skin and subsequent migration to the amputation plane (*Satoh et al., 2012*), we also investigated whether perturbing *mk* affected proliferation in the wound epidermis. Interestingly, iMDK-treated limbs displayed both consistently lower levels of wound epidermis proliferation as well as defects in the maintenance, but not induction, of blastemal cell proliferation in stump tissues (*Figure 7—figure supplement 1A,C*). However, we unexpectedly did not observe any differences in the proliferation of the wound epidermis or blastemal cells of *mk* mutants, suggesting *mk* is not required for proliferation (*Figure 7—figure supplement 1B,D*). These phenotypic differences in addition to the earlier presented blastemal cell death data were likely the main contributing causes to the observed variance in the severity in phenotype between iMDK-treated and $mk^{null}$ regenerating limbs and suggested that genetic compensatory mechanisms may be at play in the *mk* mutants (*El-Brolosy et al., 2019*). Nevertheless, overexpression of *mk* in regenerating limbs beginning at 3 dpa induced visible dramatic overgrowth of the wound epidermis by as early as 7 dpa (*Figure 7A–G*, *Figure 7—figure supplement 2*). In some *mk*-overexpressing limbs, the bone also protruded through the expanded wound epidermis, indicating that the tissue integrity of the wound epidermis was likely compromised. Further analysis confirmed that this expansion was attributed to a significant increase in the percentage of EdU$^+$ cells within the wound epidermis (*Figure 7H–I*), strongly suggesting that *mk* is sufficient, but not required, to induce proliferation in the wound epidermis.

We also observed lower levels of proliferation in blastemal cells of *mk*-overexpressing limbs (*Figure 7J*), indicating that high levels of *mk* during regeneration can have a suppressive effect on blastemal cell division. Furthermore, while control tdTomato-expressing limbs exhibited a significant decrease in monocyte density throughout regeneration, *mk*-overexpressing limbs displayed a

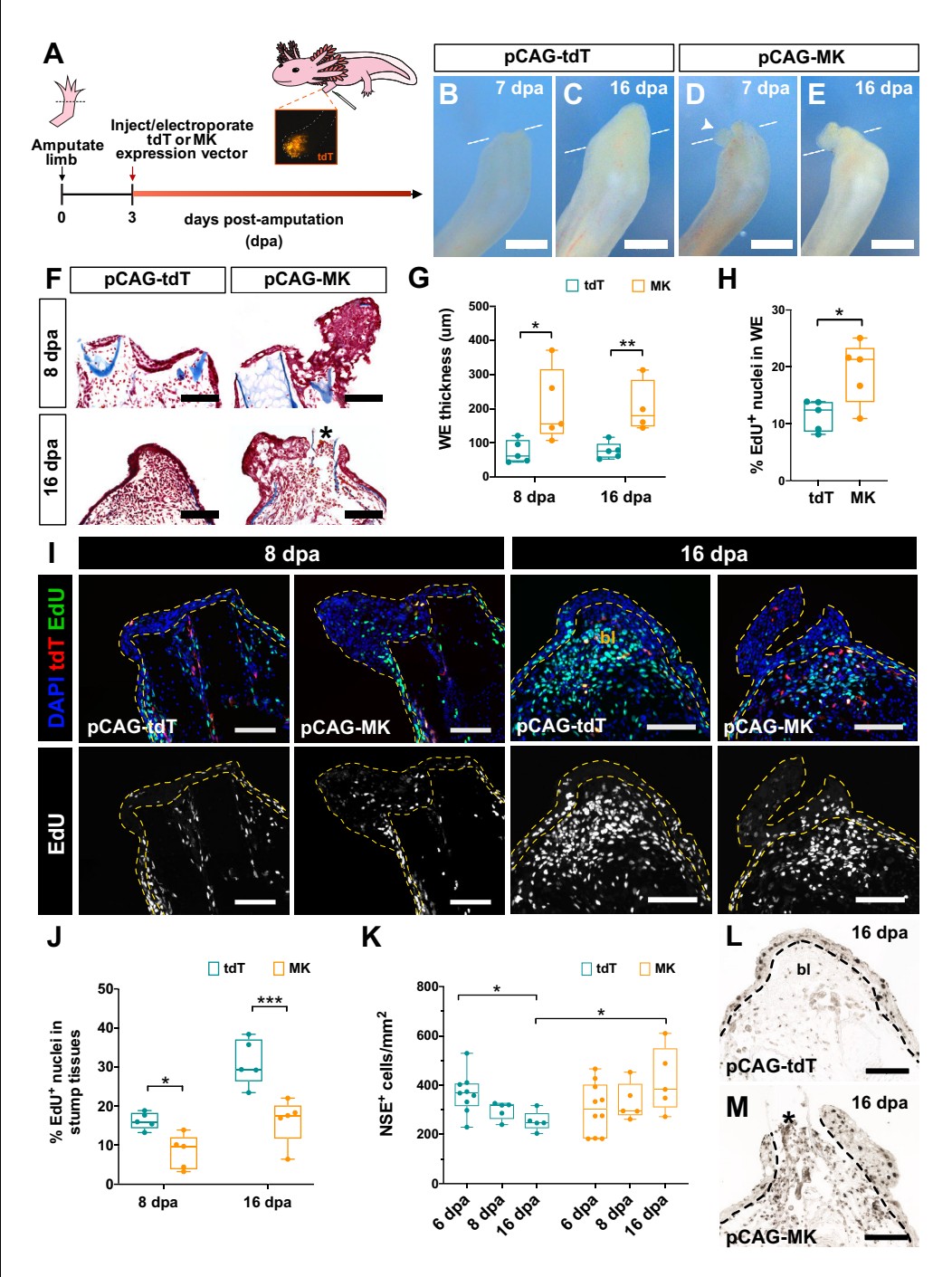

**Figure 7.** Overexpression of *mk* in regenerating limbs leads to wound epidermis expansion, decreased blastemal cell proliferation, and improper resolution of inflammation. (A) Schematic of *mk* overexpression experiment. (B–E) Representative brightfield images of tdTomato (tdT)- (B–C) or *mk*-overexpressing (D–E) regenerating limbs at 7 and 16 dpa. White dotted line denotes amputation planes and white arrowhead in D denotes small aberrant skin growth. (F) Representative picro-mallory stained sections of tdT- or *mk*-overexpressing regenerating limbs. Asterisk marks protruding bone. (G) Quantification of wound epidermis thickness in tdTomato- or *mk*-overexpressing limbs at 8 dpa (N = 5 each) and 16 dpa (N = 5 tdT, N = 4 MK). (H) Quantification of the percentage of EdU$^+$ nuclei in the wound epidermis of tdTomato- or *mk*-overexpressing limbs at 8 dpa (N = 5 each). (I) Representative images of EdU-stained sections of tdTomato- or *mk*-overexpressing limbs. Wound epidermis/AEC are outlined with yellow dotted lines. (J) Quantification of the percentage of EdU$^+$ nuclei in

*Figure 7 continued on next page*

*Figure 7 continued*

regenerating stump tissues of tdT- or *mk*-overexpressing limbs. (**K**) Quantification of the density of NSE⁺ monocytes in tdT- or *mk*-overexpressing limbs. (**L–M**) Representative images of NSE-stained sections in tdT- or *mk*-overexpressing limbs. Black dotted line denotes wound epidermis boundary. Asterisk denotes protruding bone. Quantification of EdU⁺ cells in the wound epidermis and blastemal cells of DMSO/iMDK-treated and WT/ *mk*ⁿᵘˡˡ regenerating limbs is shown in *Figure 7—figure supplement 1*. Electroporation efficiency data can be found in *Figure 7—figure supplement 2*. Data showing that *mk*-overexpressing regenerating limbs display delayed regeneration is shown in *Figure 7—figure supplement 3*. Data demonstrating that overexpression of *mk* in non-regenerating intact limbs does not affect cellular proliferation or monocyte density can be found in *Figure 7—figure supplement 4*. Graphs are mean ± SD. Two-tailed unpaired student's t-tests were used for statistical analysis. *p<0.05, **p<0.01, ***p<0.005. Scale bars, B-E: 500 µm, F, I, L-M: 200 µm. bl, blastema, dpa, days post-amputation.

The online version of this article includes the following figure supplement(s) for figure 7:

**Figure supplement 1.** *Mk* is not required for proliferation in the wound epidermis or blastemal cells.
**Figure supplement 2.** Electroporation efficiencies of pCAG-tdT and pCAG-MK injected regenerating limbs are similar.
**Figure supplement 3.** *Mk*-overexpressing limbs resolve epithelial overgrowth and undergo delayed regeneration.
**Figure supplement 4.** Overexpression of *mk* in non-regenerating intact limbs does not affect cellular proliferation or monocyte density.

persistently high monocyte density (Figure K-M), suggestive of prolonged inflammation. Together with our loss-of-function data, these results collectively suggest that *mk* levels must be tightly regulated during regeneration for proper resolution of inflammatory responses. Surprisingly, *mk*-overexpressing limbs eventually regenerated with a substantial delay, gradually resolving the overgrown wound epidermis/AEC and forming a blastema (*Figure 7—figure supplement 3*). Finally, in contrast to regeneration, overexpression of *mk* in intact limbs was not sufficient to affect cell proliferation or monocyte density (*Figure 7—figure supplement 4*), suggesting that the ability for *mk* to regulate these processes may be restricted to regenerative contexts.

## *Mk* receptors are expressed throughout regenerating limb tissues

The strong multi-tissue expression pattern of *mk* in regenerating limbs led us to investigate whether *mk* may likely be acting through autocrine or paracrine signaling. Due to *mk*'s pleiotropic functional nature, *mk* has been shown to bind to many different receptors depending on the biological context, none of which have been shown to exclusively bind *mk* (*Xu et al., 2014*). To determine the expression levels and patterns of *mk* receptors, we first examined our dataset. We found that transcriptional isoforms of several known receptors were expressed in both intact and/or early regenerating tissues (*Figure 8A*). Clustering of normalized transcript levels revealed four main clusters of expression patterns: receptors that were enriched in both the intact skin and regenerating wound epidermis/AEC, and receptors that were either downregulated, up-regulated, or maintained expression in regenerating stump tissues during regeneration.

We also examined data from published single cell limb regeneration studies and found that several of these receptors were identified as markers for various different cell types throughout homeostasis and regeneration (*Gerber et al., 2018*; *Leigh et al., 2018*). For example, *syndecans* (*sdc-1,–2, −3,–4*) and *integrins* (*integrin alpha a4, a6,* and *b1*) were detected at high levels in blastemal cells, basal and outer layers of the wound epidermis/AEC, endothelial cells, and schwann cells, indicating that *mk* likely directly signals to these cells. Surprisingly, most *mk* receptors were detected at low levels or not detected at all in myeloid cells, with the exception of *integrin b1*, which was moderately expressed (*Figure 8—figure supplement 1*). Although not a receptor, the extracellular chondroitin sulfate proteoglycan *versican* (*cspg2*) was also detected at moderate levels in myeloid cells. *Versican* is a high affinity binding partner of *mk* that can serve as an intermediary between *mk* and target receptors, potentially on immune cells (*Zou et al., 2000*; *Muramatsu, 2014*; *Wight et al., 2014*). Given the low expression of *mk* receptors in myeloid cells including monocytes/macrophages, these data altogether signify that *mk* may either directly signal through these or unidentified receptors, indirectly modulate inflammation through other tissues, or both.

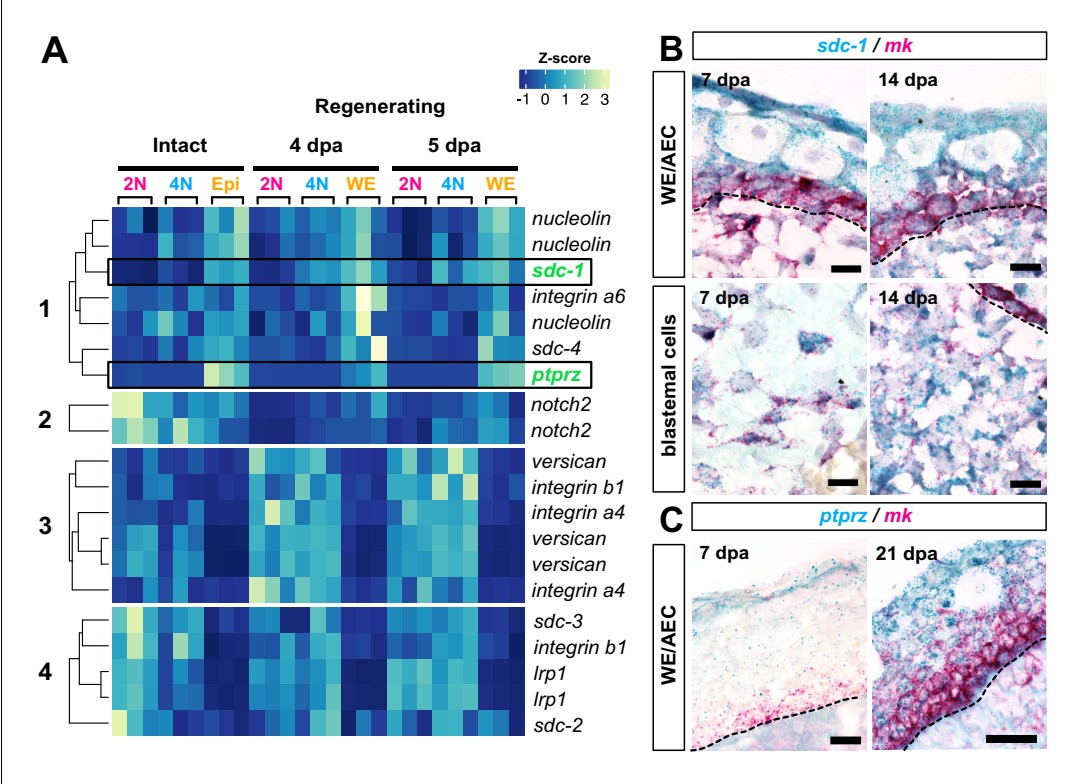

**Figure 8.** *Mk* receptors are expressed throughout regenerating tissues. (A) Heatmap of normalized transcript levels for expressed *mk* receptors in intact and regenerating subpopulations from our RNA-seq dataset revealed four general patterns of expression (*Figure 8—source data 1*). (B–C) Double RNAscope in situ hybridization of *mk* (bright red puncta) with its cognate receptors *sdc-1* (B) or *ptprz* (C) (dark blue puncta). Dotted black line denotes wound epidermis/AEC boundary. Since *ptprz* expression was low during early stages of regeneration, *ptprz*/*mk* in situs were performed without a hematoxylin counterstain to ease visualization of dark blue puncta in the wound epidermis at 7 dpa. Scale bars, B and C (left panel): 25 μm for 63x magnification, C (right panel): 50 μm for 40x magnification. WE, wound epidermis, AEC, apical epithelial cap, bl, blastema, dpa, days post-amputation. The online version of this article includes the following source data and figure supplement(s) for figure 8:

**Source data 1.** TPM values and transcript IDs for midkine receptors in *Tsai et al. (2019)*.
**Figure supplement 1.** *Mk* receptors are not highly expressed in myeloid cells during wound healing.

Interestingly, one *mk* receptor, *sdc-1* (*Mitsiadis et al., 1995b*), was expressed in both the blastemal cells and wound epidermis during early and later stages of regeneration, a pattern that seemed to overlap with that of *mk*. Double in situ hybridization of *mk* with *sdc-1* indeed validated the co-expression of *sdc-1* in *mk*+ blastemal and basal wound epidermal/AEC cells at 7 and 14 dpa (*Figure 8B*). Furthermore, *sdc-1*+ single positive cells were found in the outer layers of the wound epidermis/AEC, suggesting *mk* is capable of autocrine and paracrine signaling.

While *sdc-1* exhibited a multi-cellular expression pattern throughout regenerating tissue, we noticed that the *mk* receptor *ptprz* exhibited restricted expression in the wound epidermis/AEC (*Figure 8C*, *Figure 8—figure supplement 1*). Double in situ hybridization of *mk* and *ptprz* revealed low levels of *ptprz* expression largely restricted to the outer layers of the wound epidermis at 7 dpa. By 21 dpa, *ptprz* expression increased in the AEC and overlapped with *mk* expression in the basal layers. Furthermore, we noticed that *ptprz* failed to be up-regulated in iMDK-treated limbs, suggesting *ptprz* expression may rely on a *mk*-dependent feedback mechanism (*Figure 6C*). As signaling through *ptprz* has been shown to be important for cell survival (*Xia et al., 2019*), its restricted epithelial expression suggests that it may be a key *mk* receptor for wound epidermis-specific signaling, potentially acting in conjunction with other *mk* receptors that are also expressed in the wound epidermis. In all, these expression data demonstrate the complexity of *mk* signaling in a wide range of cell types in both limb regeneration and homeostasis.

## Discussion

While it is well known that the wound epidermis is a specialized epithelium required for the initiation of blastema formation and limb regeneration (*Goss, 1956a*; *Thornton, 1957*; *Thornton, 1958*; *Mescher, 1976*; *Tassava and Garling, 1979*), its roles during early stages of regeneration as well as the molecular mechanisms mediating its transition into the AEC remain largely unknown. Our study has revealed the functional roles of the early wound epithelium by examining how the separate transcriptional programs of blastemal progenitors and the surrounding tissues change in its absence. Here, we identified both early wound epidermis-dependent transcriptional programs as well as wound epithelial gene expression programs likely to be dependent on signaling from regenerating stump tissues. Importantly, our approach enabled us to identify *mk* as dual regulator of wound epidermis expansion during AEC development and the resolution of inflammation - two events that are essential for proper initiation of blastema formation.

The global transcriptional profiles of dividing progenitors are largely wound epidermis-independent during the initial stages of regeneration. Rather, we find that the transcriptional profiles of the surrounding tissues are more heavily affected when wound epidermis formation is prevented. A majority of these dysregulated transcripts are predominantly involved in inflammation, ECM remodeling, and tissue histolysis, indicating that the wound epidermis heavily modulates these processes during early stages of regeneration. Many characteristics of the initial wound healing stages post-amputation mirror those of mammals including activation of canonical inflammatory pathways (*Godwin et al., 2013*). Yet while most mammals activate a fibrotic scarring response post-wound healing, salamanders initiate blastema formation, suggesting there exist key differences during wound healing resolution. It is well known that differences in immune responses and ECM composition can tip the balance between scarring and regeneration (*Godwin et al., 2013*; *Godwin et al., 2014*). In light of our data, we propose that the early wound epidermis orchestrates the proper pro-regenerative niche during wound healing that is necessary for blastema formation by regulating early inflammation, ECM remodeling, and tissue histolysis.

Given that the initial cell cycle re-entry of progenitors during regeneration is independent of the wound epidermis (*Mescher, 1976*; *Tassava and Garling, 1979*; *Johnson et al., 2018*), our data provide transcriptional evidence to support the notion that the onset of progenitor activation is stimulated by processes inherent to the injury itself such as blood clotting (*Tanaka and Brockes, 1998*; *Tanaka et al., 1999*; *Wagner et al., 2017*). However, even though the overall transcriptional profile of early dividing progenitors is relatively wound epidermis-independent, we find that transcriptional activation of HIPPO, Wnt, and TGF-beta signaling in dividing progenitors is dependent on the wound epidermis. The loss of activation of not one, but all three pathways, when wound epidermis formation was prevented strongly suggests potential common regulation and synergism between these pathways, similar to other developmental and tumorigenic contexts (*McNeill and Woodgett, 2010*; *Attisano and Wrana, 2013*). As the wound epidermis/AEC is known to maintain blastemal cell proliferation (*Mescher, 1976*; *Tassava and Garling, 1979*), it is therefore possible that it does so by inducing and/or sustaining activation of HIPPO, canonical Wnt, and TGF-beta signaling in dividing progenitors. Thus, dissecting the functional interplay between these three pathways will likely elucidate the mechanisms that control early blastemal cell behaviors critical for the initial phases of blastema formation.

Common down-regulation of the macrophage marker *mpeg1* in both the non-dividing cells and epithelia of full skin flap sutured limbs further indicates that the wound epidermis may play a role in macrophage recruitment during early stages of regeneration. Macrophage depletion during early stages of limb regeneration inhibits blastema formation and results in abnormal ECM deposition reminiscent of fibrosis (*Godwin et al., 2013*), suggesting a tight interplay between the immune system and scarring. We find that many critical regenerative ECM remodeling enzymes and components, including various matrix metalloproteinases (*Vinarsky et al., 2005*) and *tenascin*, fail to be strongly induced in regenerating stump tissues in the absence of the wound epidermis. Our data therefore suggest the possibility that the wound epidermis may mediate ECM remodeling indirectly through immune cells including macrophages. Interestingly, cross-talk between the wound epithelium and macrophages through *interleukin one beta* (*il1b*) signaling is important for early stages of fish fin fold regeneration (*Hasegawa et al., 2017*). Our data also indicate that contact between epithelial cells and underlying regenerating stump tissues is important for proper regulation of

inflammation and ECM remodeling during salamander limb regeneration. In addition, we found signatures reminiscent of defects in bone resorption, indicating that the wound epidermis may play a role in mediating early skeletal remodeling. As this area has been understudied during limb regeneration, it will be interesting to explore the potential involvement of the wound epidermis in the regulation of this process in future studies.

Notably, we identify the pleotropic growth factor cytokine *mk* as a dual regulator of AEC development and the resolution of early inflammation. *Mk* is an important mediator of a variety of processes including development, tumorigenesis, and inflammation (*Muramatsu, 2010*; *Weckbach et al., 2011*; *Sorrelle et al., 2017*). We demonstrate that *mk* controls early wound epidermis expansion during AEC development by acting as a cell survival signal. Increased cell death in the wound epidermis of both iMDK-treated and *mk*^null^ regenerating limbs leads to a thin wound epithelium that exhibits aberrant gene expression and likely impaired functionality. Mis-expression of wound epidermis genes including *chs1*, which is normally only expressed during wound healing (*Leigh et al., 2018*), also demonstrates a lack of normal wound epidermis progression from early to blastemal stages. Others have shown the role of *mk* as a survival factor in several contexts including neural crest cell development, immune cell maintenance, and tumorigenesis (*Muramatsu, 2010*; *Cohen et al., 2012*; *Vieceli and Bronner, 2018*), suggesting that it may be a conserved function. In fact, *mk* signaling through one of its most well-studied receptors *ptprz* regulates cell survival (*Xu et al., 2014*). Based on our data, it is likely that *mk*-dependent *ptprz* signaling similarly modulates cell survival during AEC development in limb regeneration given its wound epidermis-specific expression pattern. Furthermore, *mk* induced anti-apoptotic signaling has been linked to activation of the PI3K/AKT signaling cascade in B-cells (*Cohen et al., 2012*), neurons (*Owada et al., 1999*), and cancer (*Dai et al., 2006*). As the inhibitor iMDK also inhibits PI3K phosphorylation (*Masui et al., 2016*), it is possible that PI3K/AKT signaling acts downstream of *mk* in AEC development. Furthermore, *ptprz* modulates endogenous phosphorylated ß-catenin levels through its phosphatase activity, which becomes inactive upon ligand binding (*Meng et al., 2000*). Since Wnt signaling is essential for limb regeneration and Wnt ligands are also expressed in the wound epidermis (*Kawakami et al., 2006*; *Yokoyama et al., 2007*; *Ghosh et al., 2008*; *Knapp et al., 2013*), *mk* may also act in concert with and/or up-stream of Wnt signaling to regulate wound epidermis development.

In a variety of other contexts, *mk* also acts as a mitogen to stimulate tumor growth, tissue growth during organogenesis, and tissue repair/regeneration (*Calinescu et al., 2009*; *Gramage et al., 2015*; *Masui et al., 2016*; *Karaman and Alitalo, 2017*; *Nagashima et al., 2019*). For example, *mk* has been shown to induce proliferation of Müller glia in an autocrine manner during zebrafish photoreceptor regeneration (*Gramage et al., 2015*; *Nagashima et al., 2019*). Therefore, we were surprised to find that *mk*^null^ mutants did not display differences in wound epidermis or blastemal cell proliferation, despite observing lower levels in iMDK-treated limbs. We also observed higher cell death in blastemal cells that eventually resolved in *mk*^null^ mutants, but not iMDK-treated limbs. In all, these data suggested that genetic compensation may be occurring in the mutant and *mk* is likely not required for general proliferation or blastemal cell survival. While it is tempting to assume compensation is the sole cause for these differences, we cannot discount the possibility that they may have been caused by non-specific effects of the inhibitor given the exact inhibitory mechanism and direct target of iMDK remains unknown (*Masui et al., 2016*). Nevertheless, while not required for proliferation, *mk* was sufficient to affect proliferation at high levels. Most notably, overexpression during regeneration induced a striking ectopic expansion of the wound epidermis. Interestingly, innervation of the wound epidermis has also been shown to regulate keratinocyte proliferation in the AEC (*Satoh et al., 2012*). While our data suggest *mk* does not affect innervation, *mk* may act in parallel with nerve-derived factors to finely control cell death/proliferation dynamics during early wound epidermis expansion. Unexpectedly, overexpression of *mk* decreased blastemal cell proliferation. Despite ample evidence of the role of *mk* as a mitogen, *mk* has been shown to have anti-proliferative activities in some contexts. For example, MK exerts an anti-proliferative effect on dental mesenchyme during mouse tooth development, yet a proliferative effect if combined with FGF-2 (*Mitsiadis et al., 1995a*), indicating that MK activity can differ depending on the presence of other growth factors in the micro-environment. Hence, *mk* may act cooperatively with other secreted molecules that modulate blastema formation and growth (*Kumar et al., 2007*; *Currie et al., 2016*;

*Farkas et al., 2016*; *Sugiura et al., 2016*; *Tanaka, 2016*; *Bryant et al., 2017b*) to exert stage- and cell-type specific effects during regeneration.

Notably, both loss- and gain-of-function of *mk* additionally led to signs of persistent inflammation including higher monocyte density, altogether suggesting that *mk* levels need to be tightly controlled for proper regulation of inflammation during early regeneration. Many relatively recent studies have focused on the pro-inflammatory role of *mk,* driving chronic inflammatory diseases including rheumatoid arthritis and atherosclerosis (*Weckbach et al., 2011*; *Sorrelle et al., 2017*). Removing or reducing *mk* activity in mouse models in these contexts led to less inflammation attributed to decreased leukocyte recruitment. In contrast, our data suggest that *mk* acts as an anti-inflammatory cytokine to resolve injury-induced inflammation during the initiation of limb regeneration. While inflammation is required for regeneration (*Kyritsis et al., 2012*; *Hasegawa et al., 2017*; *Tsarouchas et al., 2018*), prolonged inflammation can be detrimental. Recent studies have shown that macrophages play a role in resolving inflammation in both zebrafish and salamanders, by acting as a source of anti-inflammatory factors (*Godwin et al., 2013*; *Hasegawa et al., 2017*; *Tsarouchas et al., 2018*). Our data further suggest that both blastemal progenitors and the wound epidermis may be additional sources of anti-inflammatory molecules including *mk* that aid in resolving inflammation during early stages of limb regeneration. Moreover, it is possible that *mk* expression may be induced by macrophage-derived factors, an interesting future avenue of investigation. These findings thus highlight the need to advance our understanding of the complex molecular cross-talk between the immune system and various regenerating tissues during wound healing to gain insight into what leads to regenerative versus non-regenerative outcomes.

Finally, both overlapping and non-overlapping expression of *mk* and its receptors indicates that *mk* is likely acting through both autocrine and paracrine signaling in limb regeneration. The epithelial-mesenchymal dual expression of *mk* has been known for decades in the development of various organs including the lungs, gut, and limb buds of mice (*Mitsiadis et al., 1995b*). Although the expression pattern is suggestive of inductive signaling between the epithelium and developing mesenchyme, the exact functional nature of *mk* in these contexts remains largely unknown. While we did not examine *mk* expression in developing axolotl limbs, it is possible that *mk* exhibits a similar expression pattern in both the limb bud ectoderm and mesenchyme. As such, the up-regulation of *mk* in blastemal progenitors and the wound epidermis may signify the re-induction of a developmental program during the initiation of limb regeneration. Additionally, *mk* is first expressed in blastemal progenitors prior to the wound epidermis and failed to be strongly up-regulated when wound epidermis formation was prevented. Therefore, *mk* up-regulation during the initiation of limb regeneration may depend on positive feedback signaling between *mk*[+] blastemal progenitors and the wound epidermis. Moreover, *mk* receptors are expressed in intact epithelial cells and our rescue data suggest the wound epidermis is responsive to *mk* from non-epithelial sources. Hence, it is possible that *mk* secreted from blastemal progenitors may first induce *mk* expression in the wound epidermis through paracrine signaling, which is then self-sustained through autocrine signaling.

While our *mk* receptor expression data suggest that *mk* likely directly signals to the wound epidermis to regulate AEC development, how *mk* regulates inflammation remains less clear. To the best of our knowledge, besides moderate levels of *integrin b1*, we were not able to identify robust expression of *mk* receptors on populations of immune cells in our data or published single cell data. Thus, it is possible that *mk* may regulate monocyte activity directly through the low expression of these or as of yet unidentified receptor(s), indirectly through other tissues, or both. Given the data, it is likely that both regulatory mechanisms are at play. As *mk* receptors are more highly expressed throughout many other cell types and tissues, it is possible that *mk* indirectly regulates the resolution of inflammation by stimulating the secretion of anti-inflammatory factors in these tissues that then signal directly to monocytes. One likely example of such a tissue may be the wound epidermis. As we have shown that the early wound epidermis is immunomodulatory, it is conceivable that the prolonged inflammation we observed in both *mk* loss- and gain-of-function contexts was indirectly or partially caused by an impaired wound epidermis/AEC, opening up the possibility that *mk* may regulate the resolution of inflammation in an indirect manner. In all, the widespread expression of *mk* receptors throughout regenerating limb tissues underscores the complexity of *mk* function and makes it clear that future studies using tissue-specific inducible *mk* and receptor knockout models will be necessary to deconvolute the exact nature of cell-type specific *mk* signaling during regeneration.

To conclude, we have revealed the functional roles of the early wound epidermis during the initial stages of limb regeneration and further identified *mk* as a key regulator of both AEC development and the resolution of inflammation. The early expression of *mk* in both blastemal progenitors and the basal wound epidermis, key cell types that coordinate limb regeneration, suggests its expression may be linked with the ability to regenerate. As *mk* is highly conserved in other vertebrates, examining whether it plays similar roles in other appendage regeneration models and if it is differentially activated in non-regenerative systems will be important future lines of questioning. In all, we believe the findings presented here provide a solid foundation to further elucidate the molecular underpinnings of wound epidermis biology in appendage regeneration.

# Materials and methods

## Key resources table

| Reagent type (species) or resource | Designation | Source or reference | Identifiers | Additional information |
|---|---|---|---|---|
| Gene (*Ambystoma mexicanum*) | *Midkine (mk)* | | | |
| Strain, strain background (*Ambystoma mexicanum*) | Axolotl, White (d/d) | Ambystoma Genetic Stock Center (AGSC) (RRID:SCR_006372) | Cat. #AGSC_1015 | Subadults and juveniles, used for FSF transcriptional analysis and functional experiments |
| Strain, strain background (*Ambystoma mexicanum*) | Axolotl, Albino (a/a) | Ambystoma Genetic Stock Center (AGSC) (RRID:SCR_006372) | Cat. #AGSC_1025 | Subadults, used only for FSF transcriptional analysis |
| Genetic reagent (*Ambystoma mexicanum*) | $Mk^{null}$ mutants | This paper | | Generated in d/d strain |
| Cell line (*Homo sapiens*) | 293T HEK Cells | ATCC (RRID:SCR_001672) | Cat. #CRL-3216 (RRID:CVCL_0063) | For validation of MK overexpression construct |
| Antibody | Polyclonal goat anti-tdTomato | LS Bio (RRID:SCR_013414) | Cat. #LS-C340696 (RRID:AB_2819022) | WB: 1:200 |
| Antibody | Polyclonal chick anti-GAPDH | Millipore (RRID:SCR_008983) | Cat. #AB2302 (RRID:AB_10615768) | WB: 1:2000 |
| Antibody | Polyclonal rabbit anti-midkine (axolotl-specific) | This paper, NEPeptide | | IF: 1:500, WB: 1:2000 |
| Antibody | Monoclonal mouse Anti-WE3 | DSHB (RRID:SCR_013527) | Cat. #WE3 (RRID:AB_531902) | IF: 1:10 |
| Antibody | Monoclonal mouse Anti-Beta III Tubulin | Sigma Aldrich (RRID:SCR_008988) | Cat. #T8578 (RRID:AB_1841228) | IF: 1:200 |
| Recombinant DNA reagent | pCAG-tdTomato | *Pathania et al., 2012* | Addgene: Cat. #83029 | Injected and electroporated at 500 ng/uL |
| Recombinant DNA reagent | pCAG-MK | This paper | | Injected and electroporated at 500 ng/uL |
| Sequenced-based reagent | Custom RNAscope axolotl *prrx-1* probe (C1) | This paper | | In situ hybridization probe |
| Sequenced-based reagent | Custom RNAscope axolotl *csf1r* probe (C1) | This paper | | In situ hybridization probe |
| Sequenced-based reagent | Custom RNAscope axolotl *pax7* probe (C1) | This paper | | In situ hybridization probe |
| Sequenced-based reagent | Custom RNAscope axolotl *pecam* probe (C1) | This paper | | In situ hybridization probe |

*Continued on next page*

Continued

| Reagent type (species) or resource | Designation | Source or reference | Identifiers | Additional information |
|---|---|---|---|---|
| Sequenced-based reagent | Custom RNAscope axolotl *midkine* probe (C2) | This paper | | In situ hybridization probe |
| Sequenced-based reagent | Custom RNAscope axolotl *ptprz* probe (C1) | This paper | | In situ hybridization probe |
| Sequenced-based reagent | Custom RNAscope axolotl *sdc1* probe (C1) | This paper | | In situ hybridization probe |
| Sequenced-based reagent | Alt-R CRISPR-Cas9 crRNA (*mk* specific): 5'AAGCCCCCACAACTGCATCC −3' | This paper | | *Midkine* specific short guide RNA sequence |
| Sequenced-based reagent | Alt-R CRISPR-Cas9 tracrRNA | Integrated DNA Technologies (IDT) | Cat. #1072534 | |
| Sequenced-based reagent | *Mk* forward genotyping primer: 5'-TTGCTTATTCCTTGTGATCATGC-3' | This paper | | Genotyping primer |
| Sequenced-based reagent | *Mk* reverse genotyping primer: 5'- GGCACATTATTACACAGAAAGCTC-3' | This paper | | Genotyping primer |
| Sequenced-based reagent | *Mk* nested PCR forward primer: 5'- tctttccctacacgacgctcttccgatct GAGGTTTGATTGGACCCTGA-3' | This paper | | Genotyping primer |
| Sequenced-based reagent | *Mk* nested PCR reverse primer: 5'-tggagttcagacgtgtgctcttccgatct GGCACATTATTACACAGAAAGCTC-3' | This paper | | Genotyping primer |
| Peptide, recombinant protein | Axolotl Midkine blocking peptide (amino acids 126 to 142) | This paper, NEPeptide | | Used to validate custom polyclonal axolotl MK antibody |
| Chemical compound, drug | iMDK (Midkine inhibitor) | Tocris Bio (RRID:SCR_003689) | Cat. #5126 | Used at 10 uM |
| Chemical compound, drug | EdU | Thermofisher Scientific (RRID:SCR_008452) | Cat. #A10044 | Used at 8 mg/mL concentration |
| Commercial assay or kit | In Situ Cell Death Detection Kit, TMR Red | Roche (RRID:SCR_001326) | Cat. #12156792910 | |
| Commercial assay or kit | In Situ Cell Death Detection Kit, Fluorescein | Roche (RRID:SCR_001326) | Cat. #11684795910 | |
| Commercial assay or kit | Click-iT EdU Cell Proliferation Kit for Imaging | Thermofisher Scientific (RRID:SCR_008452) | Cat. #C10337 | |
| Commercial assay or kit | α-Naphthyl Acetate Esterase (NSE) Kit | Sigma Aldrich (RRID:SCR_008988) | Cat. #91A | |
| Commercial assay or kit | RNAScope 2.5 HD Duplex Assay | ACD Bio | Cat. #322430 | |
| Commercial assay or kit | Miseq Reagent Nano Kit v2 (300-cycle) | Illumina (RRID:SCR_010233) | Cat. #MS-102–2002 | |
| Commercial assay or kit | Illumina Nextera XT DNA Library Prep Kit | Illumina (RRID:SCR_010233) | Cat. #FC-131–1024 | |
| Commercial assay or kit | Ovation RNA-seq System V2 | Integrated Sciences | Cat. #7102–32 | |
| Commercial assay or kit | PrepX ILM 32i DNA Library Prep Kit | Takara Bio | Cat. #400076 | |
| Software, algorithm | Trinity | *Grabherr et al., 2011* | https://github.com/trinityrnaseq/trinityrnaseq/wiki | |
| Software, algorithm | Kallisto | *Bray et al., 2016* | https://pachterlab.github.io/kallisto/ | |

*Continued on next page*

*Continued*

| Reagent type (species) or resource | Designation | Source or reference | Identifiers | Additional information |
|---|---|---|---|---|
| Software, algorithm | Trimmomatic | *Bolger et al., 2014* | http://www.usadellab.org/cms/?page=trimmomatic | |
| Software, algorithm | DESeq2 | *Love et al., 2014* | https://bioconductor.org/packages/release/bioc/html/DESeq2.html | |
| Software, algorithm | WEB Gestalt | *Wang et al., 2017* | http://www.webgestalt.org/ | |
| Software, algorithm | CRISPResso | *Pinello et al., 2016* | http://crispresso.pinellolab.partners.org/ | |
| Software, algorithm | Complex Heatmaps | *Gu et al., 2016* | https://bioconductor.org/packages/release/bioc/html/ComplexHeatmap.html | |
| Software, algorithm | ImageJ | *Schindelin et al., 2012* | https://imagej.net/Fiji | |

## Experimental model and subject details

Axolotl (*Ambystoma mexicanum*) husbandry and surgeries were performed in accordance with the Association for Assessment and Accreditation of Laboratory Animal Care (AAALAC) and Institutional Animal Care and Use Committee (IACUC) guidelines at Harvard University. Sub-adult white and albino axolotls (15–18 cm) provided by the Ambystoma Genetic Stock Center (AGSC, University of Kentucky, KY) were used for the RNA-sequencing experiment of full skin flap sutured regenerating limbs. Juvenile white axolotls (3–6 cm animals) were utilized for the midkine inhibitor (iMDK) (Tocris Bio) experiments and *mk* overexpression experiments. *Mk* mutants were generated using CRISPR/Cas9 in the white background and genotyped/analyzed as juveniles as well.

## Animal surgeries and drug treatments

For amputations, animals were anesthetized in 0.1% tricaine (Sigma Aldrich) until unresponsive to a tail pinch. The limb was then amputated at the mid-radius/ulna level using sterilized dissection scissors. All of the limb amputations were performed on naïve regenerating animals that is animals that had not regenerated before. The amputations for all functional experiments were performed at the mid-radius/ulna level of axolotl forelimbs. For the transcriptional profiling experiment of full skin flap sutured limbs, amputations were performed at the mid-zeugopodial level (radius/ulna or –tibia/fibula) and full skin flap surgeries were performed on all four limbs of nine animals immediately post-amputation as described in *Mescher (1976)*. Briefly, limbs were amputated in the distal zeugopodial region and skin was carefully peeled back. The exposed underlying non-epithelial tissues were then amputated at the mid-zeugopodial level and the excess skin was sutured directly over the amputation plane using 9–0 sutures (Ethicon).

For iMDK experiments, limbs of animals were amputated and the animals were immersed in DMSO or 10 uM iMDK solution beginning at either 0 or 7 dpa. Drug solution was changed daily for the duration of the inhibitor experiments and animals were fed normally every other day.

## RNA isolation and sequencing

For RNA-sequencing of full skin flap sutured limbs, approximately 2–3 mm of tissue directly proximal to the amputation plane was collected at 5 dpa. Twelve limbs (6 forelimbs and 6 hindlimbs) were pooled for each replicate per timepoint and the experiment was performed in triplicate. Pooling was necessary to collect enough cells for RNA isolation.. Dissociation, DAPI cell cycle analysis, and RNA isolation from the dividing and non-dividing cells in the non-epithelial stump tissues as well as the full thickness skin flap was performed as described in *Tsai et al. (2019)*. Briefly, the wound epidermis or full skin flap at the distal amputation plane was carefully micro-dissected off of regenerating limb tissue and dissociated in 0.25% Trypsin-EDTA, while the remaining non-epithelial stump tissues along with some intact skin proximal to the amputation plane were dissociated in a mixture of collagenase, dispase, and glucose in 80% PBS. Both dissociated populations underwent DAPI staining and FACS was performed on the stained non-epithelial stump tissue fraction to sort the population

into 2N and 4N fractions. RNA was isolated from the 3 DAPI stained cellular populations of wound/full skin flap epithelial cells, dividing cells (4N) and non-dividing cells (2N) from regenerating non-epithelial stump tissues. cDNA was synthesized using the NuGEN Ovation RNAseq System V2 protocol (Integrated Sciences) using 10 ng of starting material and cDNA libraries were generated using the Wafergen PrepX Complete ILMN DNA Library kit (Takara Bio). Library concentrations were measured using the Qubit dsDNA HS Assay kit (Thermofisher Scientific) and Kapa Illumina Library Quantification Kit (Kapa Biosystems). The samples (regenerating and full skin flap samples) were pooled altogether and sequenced on an Illumina Hiseq 2500 system (Illumina) (125 bp reads) at the Harvard Bauer Core Sequencing Facility. The regenerating dataset (data from 0 and 5 dpa) presented in this study used for comparison was previously published in *Tsai et al. (2019)*, while the full skin flap samples are newly presented in this work.

Raw reads were filtered and trimmed using Trimmomatic (*Bolger et al., 2014*) and aligned to the axolotl transcriptome (*Bryant et al., 2017a*) using Kallisto (*Bray et al., 2016*). Raw TPM values from each sample were analyzed using DESeq2 (*Love et al., 2014*). Differentially expressed transcripts were filtered for fold change >2 and adjusted p-value<0.05. For pathway analysis, Uniprot IDs of blastx hits from different species for all differentially expressed transcripts were converted to human Uniprot IDs using HUGO Gene Nomenclature Committee (HGNC) and/or Uniprot ID conversion tools online. Ingenuity pathway analysis (IPA) analysis was performed on differentially expressed transcripts to determine growth factor signaling pathway activation or inhibition for the following comparisons: 0 dpa 2N vs 5 dpa 2N, 0 dpa 4N vs 5 dpa 4N, 0 dpa epi vs 5 dpa wound epidermis (WE), 0 dpa 2N vs 5 dpa FSF 2N, 0 dpa 4N vs 5 dpa FSF 4N, and 0 dpa epi vs 5 dpa FSF epi. Heatmaps of normalized TPM levels for differentially expressed transcripts in each fraction (2N, 4N, or epidermis) were generated in RStudio utilizing the ComplexHeatmap (*Gu et al., 2016*) or gplots packages (*Warnes et al., 2015*). Clusters of differentially expressed transcripts (5 dpa regenerating vs. 5dpa FSF) for each cellular fraction were analyzed using WebGestalt (*Wang et al., 2017*).

For sequencing of DMSO- or iMDK-treated limbs, RNA was isolated from DMSO or iMDK-treated regenerating limbs at 11 dpa. Tissue was homogenized in Trizol and RNA was purified using the Qiagen RNeasy mini kit. We collected the samples in biological triplicate for each condition. Quality and quantity of RNA were assessed using the Agilent RNA 6000 pico kit and run on a BioAnalyzer 2100 (Agilent). Sequencing libraries were then synthesized using the Illumina Nextera XT kit (Illumina) and sequenced on a Nextseq 500 at the Biopolymers Facility at Harvard Medical School. Fastq data were filtered, trimmed, and aligned as described above. Differential expression analysis was performed with DESeq2 (*Love et al., 2014*). For the differential expression analysis, the cutoff was set at an adjusted p-value<0.05 and fold change >2. K-means clustering of the transcripts was performed using the Complex Heatmap package (*Gu et al., 2016*) and pathway analysis of DMSO- or iMDK-enriched transcripts was performed on differentially expressed transcripts using Ingenuity Pathway Analysis (IPA) (Qiagen) software.

## Generation of mk mutants and genotyping

In order to generate the *mk* mutants, CRISPR-cas9 technology was utilized and the start codon of the *mk* locus was targeted with a single guide RNA (sgRNA) (ATG start is bolded): 5'AAGCCCCCA-CAACTGCATCC −3'. This sequence was also unique within the axolotl genome. The *mk* crRNA and generic tracrRNA were ordered from IDT and reconstituted according to their instructions. One-cell stage embryos were injected with cas9 ribonucleoprotein (cas9 RNP) complexes to generate mutants. To generate the cas9 RNP complexes, *mk* crRNA and tracrRNA were mixed and annealed to form the *mk* sgRNA. *Mk* sgRNA (final concentration 200–400 ng/μL) was then mixed with concentrated cas9 (final concentration 500 ng/μL) (PNA Bio Inc) and incubated for 10 min at 37°C to generate the cas9 ribonucleoprotein (RNP) complex. For each fertilized egg, 2–3 nl of injection mix was injected. Control embryos were injected with cas9 RNPs that were complexed with tracrRNA only.

For genotyping, genomic DNA was collected from the original limbs that were amputated from regenerating tracrRNA controls and *mk* mutants. Tissue was incubated in 50 mM NaOH for 20 min. at 94°C and neutralized with TE buffer pH 7.5. The supernatant was then PCR purified using the Qiagen PCR purification kit according to the manufacturer's instructions. Genomic PCR was performed to amplify the area surrounding the cut site with the following primers:

For 5'-TTGCTTATTCCTTGTGATCATGC-3

Rev 5'- GGCACATTATTACACAGAAAGCTC-3'

Next, two rounds of additional PCRs were performed to generate DNA sequencing libraries as described in *Gagnon et al. (2014)*. Briefly, one round of nested PCR was performed using the following gene specific primers with universal overhangs:

For 5'- tctttccctacacgacgctcttccgatctGAGGTTTGATTGGACCCTGA-3'
Rev 5'- tggagttcagacgtgtgctcttccgatctGGCACATTATTACACAGAAAGCTC-3'

The next round of PCR was performed to add on i7 indices and P5/P7 sequences to generate DNA sequencing libraries. Up to 24 libraries were pooled together and sequenced on a department owned Miseq using the Miseq reagent nano kit v2 (300 cycle) (Illumina). Fastq data were then analyzed utilizing CRISPResso (*Pinello et al., 2016*). For the genotyping analysis, mutants in which less than 1% of wildtype alleles were detected were deemed null (most had mosaic null genotypes), with up to 10–15% wildtype alleles deemed low % wildtype, and greater than 15% deemed high % wildtype (most of these animals were actually closer to 80–90% wildtype). Only mosaic null *mk* mutants were considered for the downstream characterization.

## RNAscope in situ hybridization and expression analysis

Tissue from intact or regenerating limbs was collected and fixed in 4% paraformaldehyde overnight, brought up a sucrose gradient, embedded in OCT, and cryo-sectioned at 12 μm. Chromogenic RNAscope in situ hybridization was performed utilizing the RNAscope 2.5HD Duplex Assay Kit (ACD Bio) according to the manufacturer's instructions using custom axolotl RNAscope probes generated for *mk* in the C1 channel and *pax7, prrx-1, pecam, ptprz, sdc1,* and *csf1r* in the C2 channel.

For the quantification of the breakdown of cell types expressing *mk*, sections were quantified within the distal most 500 μm from the amputation plane. The total numbers of single positive $mk^+$, $pax7^+$, $prrx-1^+$, and $pecam^+$ cells as well as double positive cells were used to calculate the percentage of $mk^+$ cells expressing each respective marker at 0, 3, and seven dpa. Although *mk* is lowly expressed in the blastema as well, the densely packed nuclei precluded accurate quantification of co-expression. For co-expression analysis of *mk* in $csf1r^+$ monocytes, single and double positive cells were counted and the respective breakdown out of total counted cells was shown.

## Histological characterization of mk mutants

Brightfield images of regenerating limbs were taken of tracrRNA controls and *mk* mutants during regeneration using an Olympus SZX16 dissecting microscope. Tissue was collected at 10 and 14 dpa for analyses. For the blastema length measurement analysis, picro-mallory staining was performed on sections and measured the length at 14 dpa. For wound epidermis thickness, the thickest part of the wound epidermis was measured and averaged across at least two sections from regenerating limbs of tracrRNA and *mk* mutants at 10 dpa.

## α-Naphthyl acetate esterase (NSE) staining and analysis

NSE staining was performed using the α-Naphthyl Acetate Esterase staining kit (Sigma) according to the manufacturer's instructions, except a 10 min fixation step was performed instead of 30 s to 1 min. The number of $NSE^+$ monocytes/unit area $mm^2$ within 500 μm of the amputation plane were quantified in DMSO/iMDK-treated, $mk^{WT}$ and $mk^{null}$, as well as tdTomato- or *mk*-overexpressing intact and regenerating limbs.

## TUNEL staining and analysis

TUNEL staining was performed using the In Situ Cell Death Detection Kit (Roche) according to the manufacturer's instructions except for tissue permeabilization which was performed by incubation at −20℃ for 5 min in 1:2 glacial acetic acid/ethanol solution. To quantify apoptotic cells in the wound epidermis, the percentage of $TUNEL^+$ nuclei out of total $DAPI^+$ cells was calculated in the wound epidermis within 500 μm of the amputation plane for each condition. To quantify the total percentage of apoptotic cells, the percentage of $TUNEL^+$ nuclei out of total $DAPI^+$ cells was calculated within 500 μm of the amputation plane.

## Immunostaining and EdU analysis

Limb tissue was collected and fixed overnight in 4% paraformaldehyde, brought up a sucrose gradient, and embedded in OCT. Tissue was cryo-sectioned at a thickness of 12 μm. For MK immunostaining, a custom polyclonal rabbit antibody against the C-terminus (amino acids 125–142) of axolotl MK (New England Peptide) was generated and validated with a peptide blocking assay. Western blots were performed on 20 μg of 10 dpa protein extracts and blocked with increasing concentrations of the peptide used to generate the antibody to validate a depletion of signal from the anti-MK staining (1:2000) (*Figure 2—figure supplement 1B*). MK immunostaining was performed at a dilution of 1:500. For other stains, the following antibodies were used: Beta-III tubulin (Sigma, 1:200), WE3 (DSHB, 1:10), GAPDH (Millipore, 1:2000), tdTomato (LS Bio, 1:200). Sections were blocked for 1 hr at room temperature (0.1% triton-X, 0.3%BSA, 8% donkey serum in PBS), incubated in primary antibodies at 4°C overnight, and stained with secondary Alexafluor antibodies.

For EdU analysis, animals were injected with 100–200 μL of an 8 mg/mL solution of EdU (Thermofisher Scientific) and the Click-it EdU A488 Imaging Kit was utilized (Thermofisher Scientific) according to the manufacturer's instructions. To quantify the proliferating cells in the wound epidermis, the percentage of EdU$^+$ cells out of total DAPI$^+$ cells in the wound epidermis was quantified within 500 μm of the amputation plane for each condition. For total EdU quantification, the percentage of EdU$^+$ cells out of total DAPI$^+$ cells across all tissues within 500 μm of the amputation plane was calculated.

## Overexpression of mk in regenerating limbs

To generate the *mk* overexpression vector, the open reading frame of *mk* was cloned into the pCAG-tdTomato backbone using EcoRI and NotI (Addgene plasmid #83029) to generate a pCAG-MK overexpression vector. To validate the expression and secretion of *mk*, 293 T cells were transfected with either pCAG-tdTomato or pCAG-MK vectors. Transfected cells and media were collected at 72 hr post-transfection (hpt). Protein lysates from the transfected 293 T cells were collected using RIPA buffer and western blots were performed on each of the protein lysates and media samples using anti-tdTomato (1:1000) (LS Bio LS-C340696), our custom generated anti-MK (1:2000), and anti-GAPDH (1:2000) (Millipore AB2302) antibodies. The presence of axolotl *mk* was detected in both pCAG-MK transfected 293 T cell lysates and media (*Figure 6—figure supplement 3A*).

Limbs of regenerating animals were injected and electroporated at 3 dpa with either the pCAG-MK overexpression vector and/or the pCAG-tdTomato vector for both the *mk* rescue experiment in *mk*$^{null}$ mutants and the general *mk* overexpression experiments. The electroporation settings were: 3 × 5 ms poring pulses at 150 V with a pulse interval of 50 ms (+ polarity, 0% decay) and 5 × 50 ms transfer pulses at 50 V with a pulse interval of 999 ms (+ polarity, 0% decay). Approximately 1 μg total was injected into each limb. Comparison of electroporation efficiency between conditions was quantified both via calculation of the mean fluorescence intensity (MFI) and quantification of the percentage of tdTomato+ cells in tissues 500 μm proximal of the amputation plane. Brightfield images of regenerating limbs were taken using an Olympus SZX16 dissecting microscope. For quantification of proliferating cells in the wound epidermis, EdU$^+$ cells were quantified out of total DAPI$^+$ cells in the wound epidermis within 200 μm of the amputation plane.

## Alcian staining

Regenerated limbs of DMSO/iMDK-treated limbs or *tdT/mk*-overexpressing limbs were collected at 60 dpa or 51 dpa, respectively, and fixed in 95% ethanol overnight. Limbs were then placed in acetone overnight and subsequently stained for 3–5 days at 37°C in Alcian blue/Alizarin red staining solution: 5% glacial acetic acid, 5% Alcian blue stock solution, and 5% Alizarin red stock solution in 70% ethanol. The tissue was then digested and cleared through a 1% KOH/glycerol gradient with increasing concentrations of glycerol.

## Mk receptor single cell RNA-seq data analysis

The single cell RNA-seq data browser was utilized from *Leigh et al. (2018)*, to identify expression patterns of *mk* receptors in different cell types during homeostasis and regeneration (https://

singlecell.broadinstitute.org/single_cell/study/SCP422/transcriptomic-landscape-of-the-blastema-niche-in-regenerating-adult-axolotl-limbs-at-single-cell-resolution-intact-limb).

### Quantification and statistical analysis

All of the imaging analysis in each functional experiment was conducted randomized and blinded to prevent bias. For statistical analysis in each experiment, an unpaired two-tailed t-test was conducted unless otherwise stated. Besides the initial full skin flap sequencing experiment, every sample (N) in each experiment was an independent biological replicate that is a limb from a different animal. All of the data were represented as the mean ± standard deviation. For quantification, the value from each biological replicate was an average of values from 2 to 3 sections from the same regions of the limb (using the radius and ulna as histological landmarks across different samples) to capture variability across the tissue.

## Acknowledgements

We thank S Ionescu and J Lavecchio in the FACS core at the Harvard Department of Stem Cell and Regenerative Biology (HSCRB) for helping with the cell sorting, I Adatto, L Krug, and Harvard Office of Animal Resources (OAR) for their dedicated animal care, as well as both the Biopolymers Facility and Bauer Sequencing Core at Harvard for their sequencing services. We are also grateful to all of the members of the Melton lab, N Leigh, J Whited, C Tabin, and C Extavour for comments and suggestions on the project and manuscript. DAM is an investigator of the Howard Hughes Medical Institute (HHMI). This research was performed using resources and/or funding from the Harvard Stem Cell Institute (HSCI) and HHMI.

## Additional information

### Funding

| Funder | Author |
| --- | --- |
| Howard Hughes Medical Institute | Douglas A Melton |

The funders had no role in study design, data collection and interpretation, or the decision to submit the work for publication.

### Author contributions

Stephanie L Tsai, Conceptualization, Data curation, Formal analysis, Validation, Investigation, Visualization, Methodology, Writing - original draft, Writing - review and editing; Clara Baselga-Garriga, Conceptualization, Data curation, Formal analysis, Validation, Investigation, Writing - review and editing; Douglas A Melton, Conceptualization, Resources, Supervision, Funding acquisition, Methodology, Writing - review and editing

### Author ORCIDs

Stephanie L Tsai (iD) https://orcid.org/0000-0001-7549-3418
Douglas A Melton (iD) https://orcid.org/0000-0002-1623-5504

### Ethics

Animal experimentation: Axolotl (Ambystoma mexicanum) husbandry and surgeries were performed in accordance with the Association for Assessment and Accreditation of Laboratory Animal Care (AAALAC) and Institutional Animal Care and Use Committee (IACUC) guidelines at Harvard University. All procedures conducted under the axolotl animal protocol #11-32 were approved by Harvard IACUC. All surgeries were performed under tricaine-induced anesthesia and every effort was made to minimize suffering.

**Decision letter and Author response**
Decision letter https://doi.org/10.7554/eLife.50765.sa1
Author response https://doi.org/10.7554/eLife.50765.sa2

## Additional files

### Supplementary files

• Supplementary file 1. Annotated differentially expressed transcripts in dividing cells (4N) non-epithelial stump tissues in full skin flap sutured vs. normal regenerating limbs. This excel table contains the list of differentially expressed transcripts in dividing cells (enriched for progenitors) in both conditions at 5 dpa. The respective fold change, blastx hit, and adjusted p-values are listed for each hit.

• Supplementary file 2. Annotated differentially expressed transcripts in non-dividing cells (2N) in non-epithelial stump tissues in full skin flap sutured vs. normal regenerating limbs. This excel table contains the list of differentially expressed transcripts in non-dividing cells in stump tissues in both conditions at 5 dpa. The respective fold change, blastx hit, and adjusted p-values are listed for each hit.

• Supplementary file 3. Annotated differentially expressed transcripts in epithelial cells of full skin flap sutured vs. normal regenerating limbs. This excel table contains the list of differentially expressed transcripts in epithelial cells of full thickness skin vs. wound epithelial cells at 5 dpa. The respective fold change, blastx hit, and adjusted p-values are listed for each hit.

• Supplementary file 4. Annotated differentially expressed transcripts in DMSO- vs. iMDK-treated regenerating limbs. This excel table contains the list of differentially expressed transcripts in DMSO vs. iMDK-treated limbs at 11 dpa with the respective fold change, blastx hit, and adjusted p-values for each hit.

• Transparent reporting form

### Data availability

The raw reads and normalized TPM values for each RNA-sequencing dataset are deposited on GEO at accession numbers GSE132325 for the full skin flap dataset and accession number GSE132317 for the iMDK dataset.

The following datasets were generated:

| Author(s) | Year | Dataset title | Dataset URL | Database and Identifier |
|---|---|---|---|---|
| Stephanie L Tsai, Clara Baselga-Garriga, Douglas A Melton | 2020 | Wound epidermis-dependent transcriptional programs | https://www.ncbi.nlm.nih.gov/geo/query/acc.cgi?acc=GSE132317 | NCBI Gene Expression Omnibus, GSE132317 |
| Stephanie L Tsai, Clara Baselga-Garriga, Douglas A Melton | 2020 | Sequencing of iMDK-treated regenerating limbs | https://www.ncbi.nlm.nih.gov/geo/query/acc.cgi?acc=GSE132325 | NCBI Gene Expression Omnibus, GSE132325 |

The following previously published dataset was used:

| Author(s) | Year | Dataset title | Dataset URL | Database and Identifier |
|---|---|---|---|---|
| Tsai SL, Baselga-Garriga C, Melton DA | 2019 | Blastemal progenitors modulate immune signaling during early limb regeneration | https://www.ncbi.nlm.nih.gov/geo/query/acc.cgi?acc=GSE111213 | NCBI Gene Expression Omnibus, GSE111213 |

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
