## [Decision Letter]

**Acceptance summary:**

Limb regeneration requires interactions between the wound epidermis and underlying stump tissues. To help clarify the role of the wound epidermis in the early stages of limb regeneration, the authors use a classic full-skin-flap suture assay, which prevents wound epidermis formation, combined with transcriptional profiling of distinct cell populations to identify genes with wound epidermis-dependent expression in the axolotl limb. Here, they focus on one such gene, encoding the growth factor midkine. Genetic ablation of midkine results in impaired limb regeneration, defects in development of the wound epidermis, elevated cell death, and persistent inflammation. Overexpression of midkine also impaired regeneration, with overgrowth of the wound epidermis, decreased proliferation in the blastema, and prolonged inflammation. This work provides an important step in understanding the mechanisms by which the wound epidermis and underlying tissues signal to each other, as well as how inflammation is regulated during regeneration.

**Decision letter after peer review:**

Thank you for sending your article entitled "Midkine is essential for wound epidermis development during the initiation of limb regeneration" for peer review at *eLife*. Your article is being evaluated by three peer reviewers, and the evaluation is being overseen by Phillip Newmark as Reviewing Editor and Anna Akhmanova as the Senior Editor.

All of the reviewers recognized the significance of this work for understanding the role of the wound epidermis in regeneration, but agreed that more evidence was required to clarify the role of midkine. If such additional evidence is provided, this work would be suitable for publication in *eLife*.

The following essential revisions should be included in a revised manuscript:

1) Grafting experiments between wild-type and midkine null animals to examine whether wild-type epithelia can rescue the defects in midkine null limbs or if midkine-null epithelia would suffice to recapitulate the null phenotype;

2) Additional consideration should be given to the effects of midkine perturbations on immune cell recruitment and blastemal cells. Does midkine overexpression lead to increases in monocytes? How do midkine perturbations affect blastemal cell survival and proliferation? Why does skin flap suture lead to a loss of macrophages while midkine inhibition leads to increased monocytes? Could looking at midkine expression during earlier time points in regeneration provide hints about the order in which it is acting in different tissues?

3) The authors suggest that the discordance between phenotypes observed after iMDK treatment and in midkine null limbs is due to genetic compensation (subsection “Mk is sufficient, but not required, for wound epidermis proliferation”), however, it seems equally (or more) plausible that these differences are due to non-specific effects of the drug. According to PubChem, high-throughput studies have identified several other targets of this drug (https://pubchem.ncbi.nlm.nih.gov/compound/15991416). At a minimum, this caveat is worth mentioning. Depending on the drug's mode of action and the level at which midkine is overexpressed, would it be possible to partially rescue drug treatment with midkine overexpression to help rule out such specificity concerns? Alternatively, are there structurally related compounds that do not inhibit midkine that can serve as better specificity controls?

To help you address their comments and clarify aspects of the manuscript that may have been unclear to the reviewers, we have included the complete reviews below.

Reviewer #1:

Tsai and colleagues build on their former transcriptional analysis (Tsai et al., 2019) with a targeted look at the role of the wound epidermis during axolotl limb regeneration. The wound epidermis has been known to be one of the essential components that allows blastema formation, although little is known about the molecular mechanisms that make it so crucial for regeneration to occur. The authors perform a classic full thickness skin flap suture over the stump tissue which prevents migration of keratinocytes and formation of the specialized wound epidermis. The wound epidermis is marked by a thickened epithelia, and importantly, no basement membrane between the epithelia and the underlying limb mesenchyme. By transcriptional profiling they find that several major signaling pathways (Wnt, TGF-B, and Hippo) are suppressed as well as a host of other differentially-expressed genes. The authors focus on midkine, a heparin-binding growth factor that has pleiotropic effects on inflammation, tumorigenesis, and tissue repair. The authors find that pharmacological and genetic perturbation can inhibit or delay limb regeneration, respectively, and loss of midkine reduces the proliferation of epithelia and mesenchymal cells while increasing apoptosis of epithelial cells and leading to a reduced thickness of the wound epidermis.

Overall, the transcriptional analysis and insights derived from the authors' work are a wonderful resource for the community and the rationale for experiments and later analysis are every bit as well done as their previous work published this year.

My primary concern is the focus on midkine as a "critical regulator of blastema formation" (taken from the impact statement). In general, I find such a conclusion to be a bit lacking given the data presented. One of the most interesting insights from the author's transcriptional analysis is that preventing wound epidermis formation primarily effects recruitment of monocytes/immune cells and their downstream activities such as activation of inflammation pathways, tissue histolysis, and ECM degradation (via MMPs). This makes me wonder why the authors decided to focus on midkine instead of the immune-epithelial crosstalk that seems to be the primary transcriptional alteration. The phenotype seen in both the skin flap suture as well as the midkine perturbations could be directly related to the loss of immune cells during regeneration. The phenotype looks very similar to the macrophage depletion described in Godwin et al., 2013, including a reduction in MMP and TGFB expression and with no loss of Twist expression.

Additionally, there are some contradictory data that should be explained -

The skin flap suture shows a loss of macrophages (by loss of mpeg1 expression) and midkine expression. However, the *mk*^null^ animals and pharmacologically-inhibited animals display higher densities of monocytes. This needs to be explained in some way, either as a technical difference in experiments or a difference in biology.

The authors propose that midkine is needed for epithelial survival and wound epidermis development. Yet, one of the most confusing aspects is the fact that midkine, a ligand, is expressed in the epithelial cells, suggesting that it may be signalling to another cell type. Or do the authors propose that midkine is signaling in an autocrine fashion? This should be explained. I also think it is a bit misleading that some of the strongest in situ signal for midkine comes from the stump tissue, which is not discussed. The high in situ signal for midkine in the stump tissue is also at timepoints (7 and 14days post amputation) that are later than the authors transcriptomic analysis (0 and 5 dpa).

The authors only briefly discuss the possibility of a role for midkine in immune cell function at the very end of the Discussion section. “Alternatively, it is also possible that *mk* may directly modulate the immune system" To my mind, this seems like the most logical conclusion that there is some immune-epithelial crosstalk mediated in part by midkine.

I think some possible revisions that could illuminate this would be:

– Does ectopic expression of midkine result in elevated monocytes or recruit immune cells? This should be done in a regenerating context as well as in intact limbs (injection of the constructs into non-regenerating tissue). Likewise, looking at monocyte numbers in the author's already-generated ectopic expression tissue (Figure 7) might give them some clues.

– Grafting experiments between *mk*^null^ and wildtype animals. Taking advantage of one of the key aspects of the axolotl system, the authors could transplant epithelial tissue between a *mk*^null^ animal and wildtype host animal (perhaps GFP+). This would inform you if 1) WT epithelia is sufficient to rescue the regeneration delay in *mk*^null^ animals, and 2) if only lacking MK expression in the epithelia can lead to the same phenotypes (limited proliferation, increased apoptosis) observed in the completely null animal. I think this could go a long way toward proving the essential nature of midkine for AEC development.

– Does ablation of macrophages by chlodronate liposomes also inhibit midkine expression similarly to skin flap suture?

– Do midkine-disrupted animals form a proper wound epidermis without a basement membrane or is it similar to the full skin flap suture? Deposition of collagens along the wound epidermis was also observed after macrophage depletion (Godwin et al., 2013) and this could link midkine and regeneration phenotype observed.

Reviewer #2:

Overall, this is a significant study which introduces Midkine as a new player in the development of the wound epidermis, a critical signaling center for salamander limb regeneration. As such, it would be of interest to researchers studying regeneration, development and related topics. Nevertheless, there are a few issues to consider prior to the publication of this work:

– In order to substantiate that the phenotypes observed following chemical and genetic *mk* perturbation are due to the specific loss of *mk* -particularly given the discrepancies of phenotypes observed between *mk*^-/-^ and iMDK-, a rescue experiment (eg overexpression of *mk* through electroporation in KO -expected to rescue- or iMDK-treated animals -expected not to rescue if the treatment is sustained-) should be conducted.

– To discard non-specific effects of CRIPR targeting, data from animals generated with a second gRNA (targeting a different site) should be presented.

– The authors show that iMDK treatment does not affect innervation, but does *mk* expression itself depend on innervation? This should be address by presenting *mk* analysis in denervated vs innervated limb blastemas.

– The authors focus on the role of MK during the expansion of the wound epidermis into the AEC, however they do not touch on the possible roles of MK itself in the blastema (notably, *prrx1^+^* cells display substantial MK expression, and loss of MK leads to increased inflammation in the entire regenerate, which could significantly impact blastema cell behaviour). This should be discussed and supported by data whenever possible. For example, EdU and TUNEL data should be shown for blastema progenitors, not just for the wound epidermis, as *mk* is also expressed in blastemal *prrx1^+^* connective tissue.

– Furthermore, they fail to explore a possible dynamic interaction between the wound epidermis and blastema cells through MK: this is evident on Figure 8, where MK receptors *sdc-1* and *ptprz* are expressed at different times in either the wep and/or blastema cells. At 7dpa, it seems that *ptprz* is expressed in blastema cells exclusively (also, *sdc-1* is expressed in blastema cells at 14dpa but not at 7dpa) and could represent an example of paracrine interaction where MK secreted by the wound epidermis affects blastema cells. Unfortunately, these issues are neither explored nor discussed, but are important in order to understand MK function.

– In this connection, it would be helpful to add additional timepoints between 1 and 3 dpa in Figure 2 in order to define whether *mk* expression comes first in the wound epidermis, in connective tissue cells, or in both at the same time.

Reviewer #3:

In this manuscript Tsai et al. use elegant techniques to identify genes expressed in the wound epidermis during regeneration. They identify midkine as a potential regulator of limb regeneration initiation.

The results are potentially interesting but I have concerns over the interpretation of some results.

1) The main results are that inhibition of midkine results in a loss of regeneration due to a loss of maintenance of proliferation. However in the CRISPR knockout the blastema still forms, it is smaller. Another possibility that is not considered is that cell division still proceeds but is slower. If the CRISPR regenerations are examined after much longer time periods does it complete regeneration?

2) In situ hybridization and immune identifies midkine to be expressed in blastema cells, which are also prxx positive, the interpretation is that 60% of the midkine positive cells are "connective tissue blastema cells". The authors need to look at other markers of cell types that contribute to the blastema like dermal fibroblasts (vimentin), satellite cells(*Pax7*). Figure 1A shows midkine expression in uninjured cells, what cells are these, maybe satellite cells?

3) Figure 7H claims that overexpression of midkine results in increased proliferation of wound epidermis cells, however in the figure it is clear that the Edu positive cells are in the blastema not the wound epidermis.

4) In what cells is midkine overexpressed? What percentage of the cells in the blastema is it overexpressed?

5) Can overexpression of midkine rescue the CRISPR knockout phenotype.

6) Are the limbs of the CRISPR knockout animal the same size as a wild type control and does the embryo develop at the same rate. A figure should be included to show these data.

7) The Discussion focuses a lot on the potential interplay between midkine and the immune system, although the RNA seq data suggests this there is very little data in the paper to support this. Does the overexpression of midkine then have an effect on the number of monocytes present or only on cell proliferation? The Discussion should be modified to reflect more the data present in the paper and then put in the broader context of regeneration

8) There is data from the zebrafish field on the role of midkine in retina regeneration this should be referred to in the Discussion.

9) In general the N for a lot of experiments is very low, in some animal experiments the N=4, which means the data are not statistically relevant. The n should be increased to a minimum of 10 for all animal experiments.

10) The Material and methods section is incomplete, RNAscope is used for several figure but there is no methods for it included.

---

## [Author Response]

Reviewer #1:[…]I think some possible revisions that could illuminate this would be-– Does ectopic expression of midkine result in elevated monocytes or recruit immune cells? This should be done in a regenerating context as well as in intact limbs (injection of the constructs into non-regenerating tissue). Likewise, looking at monocyte numbers in the author's already-generated ectopic expression tissue (Figure 7) might give them some clues.

We thank the reviewer for this suggestion. We performed additional experiments to address this point. We found that ectopic expression of midkine in a regenerating context also led to an elevated density of monocytes. While control tdT-overexpressing limbs exhibited a significant decrease in the number of monocytes, from 6 -16 dpa, *mk*-overexpressing limbs showed consistently high level of monocytes, indicating aberrantly high levels of *mk* can induce persistent inflammation. These additional data have now been included in Figure 7K-M. We also performed *mk* overexpression experiments in intact limbs and saw no difference in proliferation or monocyte density, suggesting *mk* overexpression in this context is not sufficient to regulate these processes during homeostasis. The data from these experiments are described in the manuscript and presented in Figure 7—figure supplement 4. Altogether, these data provide supportive evidence that *mk* levels must be tightly regulated for proper resolution of inflammation during regeneration.

– Grafting experiments between mk-^null^ and wildtype animals. Taking advantage of one of the key aspects of the axolotl system, the authors could transplant epithelial tissue between a mk-^null^ animal and wildtype host animal (perhaps GFP+). This would inform you if 1) WT epithelia is sufficient to rescue the regeneration delay in mk-^null^ animals, and 2) if only lacking MK expression in the epithelia can lead to the same phenotypes (limited proliferation, increased apoptosis) observed in the completely null animal. I think this could go a long way toward proving the essential nature of midkine for AEC development.

We thank the reviewers for proposing this elegant experiment and agree that whether the presence of *mk* in the epithelia alone is sufficient to rescue the phenotype and vice versa is an interesting and important question. On this note, we had originally thought about conducting this exact experiment, however we decided for several reasons that the results from grafting experiments would be confounding. Grafting just the wound epidermis alone is technically difficult and has a very low success rate. To perform this experiment in the most reproducible manner, we would instead graft a cuff of full thickness skin from a *mk*^null^ mutant to a wildtype intact limb and vice versa, then amputate through the graft and observe regeneration. Grafting full thickness skin would also include *prrx1^+^* dermal fibroblasts that highly express *mk* and are major contributors to the blastema (Kragl et al., 2009, Gerber et al., 2018). Therefore, if we observed normal wound epidermis expansion when grafting wildtype epithelia onto mutant limbs, we could not conclude that the rescue in phenotype was due to the presence of *mk* in only the epithelia. While performing this grafting experiment could not directly address the question posed, we did agree that this was an important issue. Therefore, we asked the converse question of whether *mk* derived from stump tissues is sufficient to rescue the mutant phenotypes. To do this, we exploited the fact that electroporating constructs into the limb epidermis is difficult and only occurs at extremely low efficiency (Fei et al., 2016). We performed a rescue experiment, in which we overexpressed either tdTomato (as a control) and/or *mk* in *mk*^null^ mutants. As expected, the majority of *mk* was expressed and secreted from regenerating stump tissues as can be seen in Figure 6—figure supplement 3. Overexpression of *mk* in *mk*^null^ mutants was sufficient to rescue the delayed regeneration, AEC developmental defects (thin wound epidermis/high cell death), and high monocyte density. These data altogether suggest 1) that the wound epidermis is responsive to *mk* secreted from non-epithelial sources and 2) that *mk* from stump tissues is capable of signaling to the wound epidermis and likely vice versa. These additional data can be found in Figure 6—figure supplements 1 and 2.

– Does ablation of macrophages by chlodronate liposomes also inhibit midkine expression similarly to skin flap suture?

This is an interesting question to explore. In response, we attempted a small pilot experiment in a collaborative effort with Dr. Nicholas Leigh, a postdoctoral fellow in the Whited lab. The original dosage in Godwin et al., 2013, used 12.5 mg/kg and we therefore bracketed this using 2x, 1x, 0.5x mg/kg. Each condition had three animals (1 control, 2 experimental). Unfortunately, in our hands, treatment with clodronate caused severe edema at all three doses which we presumed was due to cytotoxicity associated with the clodronate and/or opportunistic infections in both the regenerating and intact hindlimbs of depleted animals. As a result, many animals had to be euthanized. These cytotoxic effects precluded the interpretation of the results on *mk* expression (data not shown) and the optimization of this method extends beyond the scope of the current manuscript. However, we do agree this is a possibility and have added this as a future direction in the Discussion (paragraph seven).

– Do midkine-disrupted animals form a proper wound epidermis without a basement membrane or is it similar to the full skin flap suture? Deposition of collagens along the wound epidermis was also observed after macrophage depletion (Godwin et al., 2013) and this could link midkine and regeneration phenotype observed.

Picro-mallory staining of the wound epidermis/AEC in both iMDK- and *mk*^null^ mutants indicates a lack of a basement membrane similar to a normal wound epidermis (Figure 4D’ and G’, absence of blue collagen staining at the basal surface of the wound epidermis). On the other hand, picro-mallory staining of the full skin flap sutured limbs shows a clear blue collagen thick basement membrane and dermal layer (Figure 1—figure supplement 1B’). Therefore, *mk*-disrupted animals appear to form a proper wound epidermis with regards to this aspect. Furthermore, the basement membrane and blue collagen thick dermal layer could be seen re-forming at the amputation plane in iMDK-treated limbs much later at 40 dpa (now included in Figure 4—figure supplement 1), providing further evidence for its absence during earlier stages when AEC development normally occurs.

Reviewer #2:– In order to substantiate that the phenotypes observed following chemical and genetic mk perturbation are due to the specific loss of mk -particularly given the discrepancies of phenotypes observed between mk-/- and iMDK-, a rescue experiment (eg overexpression of mk through electroporation in KO -expected to rescue- or iMDK-treated animals -expected not to rescue if the treatment is sustained-) should be conducted.

We thank the reviewer for this suggestion and agree that this is an important point to address given the difference in phenotypes between iMDK treatment and *mk*^null^ mutants. As such, we performed a rescue experiment in which we overexpressed *mk* in *mk*^null^ mutants beginning at 3 dpa (the time point when *mk* exhibits strong induction). Overexpression of *mk*, but not tdTomato (as a control), in *mk*^null^ mutants was sufficient to rescue the associated phenotypes including delayed regeneration, defective AEC development (thin wound epidermis and higher cell death), as well as high monocyte density associated with persistent inflammation. We believe that these additional data (which can be found in Figure 6—figure supplements 1, 2 and 3) provide strong evidence that *mk* indeed regulates both AEC development and the resolution of inflammation.

– To discard non-specific effects of CRIPR targeting, data from animals generated with a second gRNA (targeting a different site) should be presented.

While targeting *mk* with a second gRNA against a different site would provide additional evidence to reinforce our conclusions, we believe that the correlation of phenotypes between two independent loss-of-function methods (chemical and genetic) in our study provides sufficient evidence for our claims. Additionally, our *mk* rescue experiment in the mutant further supports our conclusions that the phenotypes we observed were indeed due to *mk* deficiency and not off-target effects of CRISPR. Together, we believe these data are sufficient to support our conclusions that *mk* is a regulator of AEC development and inflammation.

– The authors show that iMDK treatment does not affect innervation, but does mk expression itself depend on innervation? This should be address by presenting mk analysis in denervated vs innervated limb blastemas.

We thank the reviewer for raising this interesting point. We have taken advantage of a previous published microarray dataset in both innervated and denervated regenerating axolotl limbs to answer this question. The data from this study determined that *mk* was up-regulated at a similar fold change in both innervated and denervated limbs indicating its upregulation is independent of innervation. We have now included this commentary in the Results (paragraph three in subsection “Mk loss-of-function leads to impaired AEC development and persistent inflammation”) along with referencing the appropriate study (Monaghan et al., 2009).

– The authors focus on the role of MK during the expansion of the wound epidermis into the AEC, however they do not touch on the possible roles of MK itself in the blastema (notably, prrx1^+^ cells display substantial MK expression, and loss of MK leads to increased inflammation in the entire regenerate, which could significantly impact blastema cell behaviour). This should be discussed and supported by data whenever possible. For example, EdU and TUNEL data should be shown for blastema progenitors, not just for the wound epidermis, as mk is also expressed in blastemal prrx1^+^ connective tissue.

We agree with the reviewer on this point and have conducted additional analyses on blastemal cell death and proliferation in iMDK-treated and *mk*^null^ regenerating limbs. While we did observe decreased proliferation and persistently higher levels of cell death in blastemal cells of iMDK-treated limbs, we did not observe differences in blastemal cell proliferation or death in *mk*^null^ mutants. These experimental data are shown in Figure 5—figure supplement 1 and Figure 7—figure supplement 1. Together, these data suggest that *mk* is not required for blastemal cell proliferation or cell survival. Alternatively, the discrepancies in the phenotypes could be attributed to off target effects of the inhibitor, which we state and elaborate on in the Discussion (paragraph six).

– Furthermore, they fail to explore a possible dynamic interaction between the wound epidermis and blastema cells through MK: this is evident on Figure 8, where MK receptors sdc-1 and ptprz are expressed at different times in either the wep and/or blastema cells. At 7dpa, it seems that ptprz is expressed in blastema cells exclusively (also, sdc-1 is expressed in blastema cells at 14dpa but not at 7dpa) and could represent an example of paracrine interaction where MK secreted by the wound epidermis affects blastema cells. Unfortunately, these issues are neither explored nor discussed, but are important in order to understand MK function.

We apologize if the initial *ptprz* in situ at 7 dpa was misleading. The strong blue signal shown in the stump tissues was non-specific background staining from the fibrin clot at the amputation plane. We have included new in situs at 7 dpa for *ptprz* that more clearly demonstrate low expression of *ptprz* (dark blue puncta) throughout the wound epidermis with minimal background staining in the clot (Figure 8C). This was also validated by single cell RNA-seq expression data from Leigh et al., 2018 (Figure 8—figure supplement 1). That said, we thank the reviewer for bringing this to our attention and agree that a more extensive analysis and discussion of the potential signaling mechanisms of *mk* was needed to put the findings in this study into a broader regenerative context.

We have now included a more comprehensive analysis of the expression of midkine receptors in our dataset as well as publicly available single cell data (Figure 8 and Figure 8—figure supplement 1). We find overlapping and non-overlapping expression patterns for *mk* receptors and *mk*, indicating *mk* is capable of both autocrine and paracrine signaling. Furthermore, we find the expression of multiple *mk* receptors in the wound epidermis, indicating *mk* likely directly binds to and regulates wound epidermis/AEC development. However, by mining the higher resolution single cell data, we were surprised to find that *mk* receptors were largely detected at low levels or not detected at all in myeloid cell populations, with the exception of integrin β 1. These data bring forth the possibilities that *mk* may: 1) directly regulate myeloid cell populations through the low expression of these or as of yet unidentified receptors, 2) indirectly regulate inflammation via signaling through other tissues, or 3) modulate inflammation through both. Given the *mk* receptor expression data, we speculate in the Discussion that *mk* is likely regulating the resolution of inflammation through both mechanisms and elaborate on how our functional and transcriptional data suggest that the wound epidermis may be an example of one tissue that may indirectly mediate resolution of inflammation.

– In this connection, it would be helpful to add additional timepoints between 1 and 3 dpa in Figure 2 in order to define whether mk expression comes first in the wound epidermis, in connective tissue cells, or in both at the same time.

We agree with the reviewer that a more in-depth analysis of the onset of *mk* expression in regenerating tissues would provide important clues as to the dynamics of *mk* signaling. In response, we have now included an additional timepoint, 56 hours post-amputation (hpa), and high magnification images of *mk* expression in both the wound epidermis and stump tissues at 24, 56, and 72 hpa demonstrating that *mk* is first expressed in cells within regenerating stump tissues prior to the wound epidermis (Figure 2—figure supplement 2A). These data therefore suggest that *mk* expression in the wound epidermis may first be induced through paracrine signaling from *mk* and/or other extracellular molecules in regenerating stump tissues.

Reviewer #3:1) The main results are that inhibition of midkine results in a loss of regeneration due to a loss of maintenance of proliferation. However in the CRISPR knockout the blastema still forms, it is smaller. Another possibility that is not considered is that cell division still proceeds but is slower. If the CRISPR regeneration are examined after much longer time periods does it complete regeneration?

We agree with the reviewer on this point and now include data indicating that the *mk*^null^ mutants do not exhibit defects in blastemal cell proliferation, indicating *mk* is likely not required for blastemal cell division. The *mk*^null^ mutants do regenerate anatomically normal limbs, although delayed and we have now included time-lapse imaging in Figure 6—figure supplement 1 to demonstrate the delay in mutants compared to wildtype and *mk*-rescued animals.

2) In situ hybridization and immune identifies midkine to be expressed in blastema cells, which are also prxx positive, the interpretation is that 60% of the midkine positive cells are "connective tissue blastema cells". The authors need to look at other markers of cell types that contribute to the blastema like dermal fibroblasts (vimentin), satellite cells(Pax7). Figure 1A shows midkine expression in uninjured cells, what cells are these, maybe satellite cells?

We thank the reviewer for raising this issue and have conducted a more detailed double in situ hybridization analysis to identify the cell types that express *mk* during homeostasis and regeneration. The analysis shows that the majority of *mk*-expressing cells are *pax7^+^* satellite cells and *pecam^+^* endothelial cells, in addition to *prrx1^+^* connective tissue cells, in both intact and regenerating tissues. We quantified the breakdown of cell types expressing *mk* in regenerating stump tissues and included representative images of *mk* double in situs in both intact and regenerating tissues (3, 7, and 24 dpa). In intact tissues, connective tissue cells, satellite cells, and endothelial cells comprised approximately 43.8%, 44.9% and 7.7% of *mk*-expressing cells in stump tissues, respectively. During regeneration, the majority of *mk*-expressing cells were connective tissue cells (~53-54%) and satellite cells (~36-37%) at 3 and 7 dpa. A smaller percentage (~4-5%) of *mk*-expressing cells were endothelial cells at both time points. These data are now included in the study and can be found in Figure 2H-I and in Figure 2—figure supplement 2B.

3) Figure 7H claims that overexpression of midkine results in increased proliferation of wound epidermis cells, however in the figure it is clear that the Edu positive cells are in the blastema not the wound epidermis.

We apologize if this was unclear in the figure. The EdU+ cells are indeed in the wound epidermis as outlined in the dotted yellow lines in each panel of Figure 7I. At 8 dpa, a blastema has not formed in both tdT- and *mk*-overexpressing limbs (this time point is pre-blastema formation). Furthermore, additional quantification of EdU+ cells in the regenerating stump tissues revealed a significant decrease, not increase in proliferation of blastemal cells at 8 and 16 dpa. The additional data quantifying proliferation in blastemal cells is shown in Figure 7J.

4) In what cells is midkine overexpressed? What percentage of the cells in the blastema is it overexpressed?

Overexpression of constructs via electroporation in axolotl limbs predominantly results in expression in stump tissues, as the electroporation into the epithelial tissues is difficult and occurs at exceedingly low efficiencies. The most obvious expression can be seen in myofibers, however other cell types can also uptake constructs through electroporation (Fei et al., 2016). We have included representative images of tdTomato staining in both our *mk* overexpression experiment during regeneration and our newly added *mk* rescue experiment in Figure 7—figure supplement 2 and Figure 6—figure supplement 3, respectively. In both experiments, quantification showed that up to 20% of cells constitutively expressed tdTomato or *mk*. We also show equivalent efficiencies by both cell quantification and mean fluorescence intensity across control and experimental conditions as supportive control data to discount the possibility that the phenotypic effects were due to varying electroporation efficiencies.

5) Can overexpression of midkine rescue the CRISPR knockout phenotype.

We thank the reviewer for raising this important point and have now included data indicating that overexpression of *mk* beginning at 3 dpa in our *mk*^null^ mutants is sufficient to rescue the delayed regeneration, AEC developmental defects (thin wound epidermis and higher cell death), as well as higher monocyte density. These data can be found in Figure 6—figure supplement 1, 2 and 3.

6) Are the limbs of the CRISPR knockout animal the same size as a wild type control and does the embryo develop at the same rate. A figure should be included to show these data.

We have now included representative images of *mk*WT and *mk*^null^ mutants indicating no visible difference in size of the animals as well as anatomically normal limbs. These data can be found in Figure 3—figure supplement 1.

7) The Discussion focuses a lot on the potential interplay between midkine and the immune system, although the RNA seq data suggests this there is very little data in the paper to support this. Does the overexpression of midkine then have an effect on the number of monocytes present or only on cell proliferation? The Discussion should be modified to reflect more the data present in the paper and then put in the broader context of regeneration

We agree that additional evidence is needed to support these claims. As such, we now include data showing that overexpression of *mk* during regeneration leads to consistently higher monocyte density compared to control limbs, indicative of chronic inflammation (Figure 7K-M). Together with the loss-of-function data, these data show that mk levels need to be tightly maintained to proper modulate the resolution of inflammation. We also present additional data showing that *mk* overexpression leads to decreased blastemal cell proliferation despite seeing the opposite effects in the wound epidermis (Figure 7J). We have expanded the Discussion section to incorporate all of these data into the broader context of the roles of *mk* in regeneration.

8) There is data from the zebrafish field on the role of midkine in retina regeneration this should be referred to in the Discussion.

We thank the reviewer for bringing this to our attention and have now included the relevant references in the Discussion (paragraph six) (Gramage et al., 2015, Nagashima et al., 2019).

9) In general the N for a lot of experiments is very low, in some animal experiments the N=4, which means the data are not statistically relevant. The n should be increased to a minimum of 10 for all animal experiments.

While the reviewer brings up a valid and important point, we note that similar statistically significant effects are observed across two different loss-of-function methods and corroborative gain of- function experimental evidence for the role of *mk* in both wound epidermis development and inflammation. Moreover, the results from the *mk* rescue experiment in the mutants provide new additional data to support our claims. The statistical methods utilized are included in the legend of each figure as well as the Quantification and Statistical Analysis section of the Materials and methods.

10) The Materials and methods section is incomplete, RNAscope is used for several figure but there is no methods for it included.

We thank the reviewer for pointing this out and have now included a detailed section in the Materials and methods on RNAscope.